

# Assessment of the ParFlow-CLM CONUS 1.0 integrated hydrologic model: Evaluation of hyper-resolution water balance components across the contiguous United States

Mary M. F. O'Neill[1,2,3], Danielle T. Tijerina[1,4], Laura E. Condon[5], Reed M. Maxwell[1,4]

[1]Colorado School of Mines, Department of Geology and Geological Engineering, Golden, CO, USA

[2]Now at NASA Goddard Space Flight Center, Hydrological Sciences Laboratory, Greenbelt, MD, USA

[3]Now at University of Maryland, College Park, Earth System Science Interdisciplinary Center, Greenbelt, MD, USA

[4]Now at Princeton University, Department of Civil and Environmental Engineering and Princeton Environmental Institute, Princeton, NJ, USA

[5]The University of Arizona, Department of Hydrology and Atmospheric Sciences, Tuscon, AZ, USA

**Abstract**

Recent advancements in computational efficiency and earth system modeling have awarded hydrologists with increasingly high-resolution models of terrestrial hydrology, which are paramount to understanding and predicting complex fluxes of moisture and energy. Continental-scale hydrologic simulations are, in particular, of interest to the hydrologic community for numerous societal, scientific and operational benefits. The coupled hydrology-land surface model ParFlow-CLM configured over the continental United States (PFCONUS) has been employed in previous literature to study scale-dependent connections between water table depth, topography, recharge, and evapotranspiration, as well as to explore impacts of anthropogenic aquifer depletion to the water and energy balance. These studies have allowed for an unprecedented, process-based understanding of the continental water cycle at high resolution. Here, we provide the most comprehensive evaluation of PFCONUS version 1.0 (PFCONUSv1) performance to date, comparing numerous modeled water balance components with thousands of in situ observations and several remote sensing products, and using a range of statistical performance metrics for evaluation. PFCONUSv1 comparisons with these datasets are a promising indicator of model fidelity and ability to appropriately reproduce the continental-scale water balance at high resolution. Areas for improvement are identified, such as a positive streamflow bias at gauges in the eastern Great Plains, a shallow water table bias over many areas of the model domain, and low bias in seasonal total water storage amplitude especially for the Ohio, Missouri and Arkansas river basins. We discuss several potential sources for model bias and suggest that minimizing error in topographic processing and meteorological forcing would considerably improve model performance. Results here provide a benchmark and guidance for further PFCONUS model development, and they highlight the importance of concurrently evaluating all hydrologic components and fluxes to provide a multivariate, holistic validation of the complete modeled water balance.



# 1 Introduction

Explicitly modeling the terrestrial water cycle at the global scale and at high resolution has recently been referred to as a "grand challenge in hydrology" (Bierkens et al., 2015; Wood et al., 2011), an undertaking that has excited the hydrologic community and encouraged the development of large-scale modeling efforts, workshops and

working groups. These "everywhere and locally relevant" hydrologic models (Bierkens et al., 2015) differ from land surface models (LSMs) and general circulation models (GCMs), by providing spatially ubiquitous and hyper-resolution, physically-based hydrologic simulations. While LSMs and GCMs may provide water balance estimates at regional, continental or global scales, their hydrologic schemes can be coarse resolution, simplified or highly parameterized (Wood et al., 2011). A process-based and mechanistic (rather than empirical) representation of both

the large-scale and local water cycle is necessary to address hydrologic problems surrounding society, agriculture, resource management, biodiversity, and climate (Clark et al., 2015).

Therefore, high-resolution, large-scale and physically based hydrologic modeling offers profound and multi-faceted benefits. From a societal perspective, these models enable operational forecasting and planning in regions where water balance estimates are unavailable or poorly constrained by scarce or nonexistent observations,

such as developing countries (Group on Earth Observations, 2009). As Beven and Cloke (2012) point out, hyper-resolution hydrologic model outputs (as opposed to course-resolution global hydrologic model (GHM) results) can be more accessible and logical to local water managers by providing locally relevant and detailed information. High-resolution hydrologic modeling could also be used to inform, initialize, or downscale LSMs and GCMs. Clark et al. (2015) identify spatial heterogeneity, organization and integration of soil moisture and groundwater to be a major

missing link in LSMs, meteorological models and climate models. Further, large-scale hydrologic models could be used to better understand or constrain results from remote sensing. For instance, LSMs may be used in forward modeling approaches to estimate signal attenuation in remote sensing of total water storage change (Landerer & Swenson, 2012).

These motivating factors have catalyzed the development of several hyper-resolution, continental- or

global-scale modeling efforts over the last decade. Some fine examples include physically based platforms, such as the Terrestrial Systems Modeling Platform (TerrSysMP), a fully integrated soil-vegetation-atmosphere model, employed over the European CORDEX domain (Keune et al., 2016); and integrated groundwater-surface water modeling over the continental United States with ParFlow v3 (Maxwell et al., 2015). Others have used a global water balance approach, like WaterGAP (Döll et al., 2003), as well as PCR-GLOBWB (Sutanudjaja et al., 2018),

which was recently coupled to MODFLOW at globally 1-km resolution (de Graaf et al., 2017). High-resolution land surface modeling has begun to include topographically informed routing of surface or subsurface water storage; for example, the Land Information System software group (Zaitchik et al., 2010) or Noah-MP (Niu et al., 2011); and operational flood forecasting from the National Water Model (NWM) v2.0 (*Office of Water Prediction*, water.noaa.gov/about/nwm). Many of these platforms were made possible given the notable progress made in

globally available and openly accessible input parameters, such as hydrography datasets (e.g. Lehner et al., 2008) and hydraulic parameters (e.g., Survey, 2003; BGR & Unesco, Groundwater Resources of the World, n.d.; Gleeson



et al., 2014; SSURGO), as well as advancements in computational efficiency and massively parallel computing resources (e.g., Kollet et al., 2010).

While global and continental hydrologic representation continues to improve, the extreme-scale hydrologic modeling community still faces many challenges and models can struggle to close the water balance with certainty. Given the lack of spatially and temporally continuous hydrologic measurements across the globe, as well as their associated computational demand, parameter calibration at these scales is often problematic or infeasible (R. M. Maxwell et al., 2015). Distributed macroscale hydrology models must often rely on a priori information and datasets informed by field measurements or hydrologic theory (which may be unavailable, especially in under developed

regions); or, less commonly, they can employ regionalization approaches to transfer calibrated parameters from gauged to ungauged catchments (Beck et al., 2016). Validation can also be problematic, in that large gaps exist in space or time for in situ measurements, and remote sensing products often depend on hydrologic algorithms and parameterization (Archfield et al., 2015).

Studies assessing model performance suggest that while continental and global hydrologic modeling is

promising, there is considerable room for improvement when it comes to model skill, and most of these performance assessments only evaluate one or two output variables at one time. For instance, Sutanudjaja et al. (2018) evaluated streamflow and total water storage performance of 5 arcmin resolution simulation of PCR-GLOBWB relative to the Global Runoff Data Center (GRDC) and remote sensing from the Gravity Recovery and Climate Experiment (GRACE). Although TWS performance was generally acceptable for major global river basins, they found that only

40% of discharge locations exhibited a Kling-Gupta efficiency coefficient (KGE, a measure of performance in which 0.5 or lower is unsatisfactory; Bai et al., 2016; Moriasi et al., 1983) of >0.3, suggesting that the large majority of GRDC stations show unsatisfactory performance. Recent streamflow results from WaterGAP2.2d are encouraging (Schmied et al., 2020), with a median KGE of 0.79 and a near-optimum bias measure; however, the model underestimated TWS amplitude and trend in the majority of basins. Salas et al. (2018) evaluated the National

Flood Interoperability Experiment (NFIE-Hydro), which leverages the WRF-Hydro framework and the Noah-MP LSM. They identify several regions for model improvement, including a positive bias of flow in the Southern U.S and Central Plains and a negative bias in the Rocky Mountains, suggesting several potential sources for bias depending on the area, including snowpack formulation, precipitation bias, soil column draining dynamics, or failure of lateral redistribution to attenuate flow. These results reiterate that acceptable performance of one model output

does not necessarily translate to appropriate simulations of the full water balance, and evaluating multiple output parameters simultaneously (such as snow water equivalent, soil moisture, evapotranspiration, and many others) could help confidently attribute sources of bias.

We argue that validation and performance assessment should continue to be highly prioritized for uncalibrated, high-resolution, and large-scale hydrologic models, and validation studies that evaluate several output

variables are paramount to guiding and improving model development. It has been well established that calibration methods utilizing multiple types of observational datasets result in overall better model skill (e.g., Finger et al., 2015); additionally, understanding the relationships between multiple output variables (e.g., evaporative fraction and soil saturation, Rakovec et al., 2019) is imperative to diagnosing performance deficiencies. Multivariate model





validation can help attribute sources of bias and increase certainty in water balance components; this is especially true for the physically based hydrologic community and at continental scales and above.

In this study, we present a rigorous, multivariable evaluation of a hyper-resolution continental-scale hydrologic simulation, comparing model results to state-of-the-art monitoring networks and remote sensing products. We focus on performance of the CONUS version 1.0 model, a ParFlow-CLM integrated groundwater-surface water simulations configured across the continental United States (hereby referred to as PFCONUSv1) (Maxwell et al., 2015). Since its construction, the PFCONUSv1 model has been updated to a ParFlow-CLM simulation, in which ParFlow is coupled to the Common Land Model to capture surface energy partitioning and land surface fluxes (Maxwell & Miller, 2005). Recent publications have used the PFCONUSv1 model to 1) diagnose mechanistic relationships between water table depth, topography, recharge and evapotranspiration at a range of scales (Condon et al., 2015; Condon & Maxwell, 2015, 2017); 2) characterize groundwater controls on evapotranspiration partitioning (Maxwell & Condon, 2016); 3) explore anthropogenic impacts to the water and energy balances, such as impacts to evapotranspiration, streamflow and groundwater from aquifer depletion (Condon et al., 2020; Condon & Maxwell, 2019); and 4) estimate water residence times and their sensitivity to climate and geology (Maxwell et al., 2016).

To our knowledge, this is the most rigorous evaluation of an integrated, physically based hydrology-land surface model at this resolution and scale. We present comparisons of model results and observations or remote sensing products over four simulation years (water years 2003 through 2006) for several water balance components, including streamflow, water table depth, soil moisture, snow water equivalent, evapotranspiration, and total water storage, as well as atmospheric forcing (precipitation and temperature). We discuss sources of error in the model and prioritize areas for improvement, with careful attention to error propagation from atmospheric forcing datasets and terrain processing algorithms. These results provide a benchmark for forthcoming PFCONUS iterations and should be used to guide future model development. Most importantly, this study implicates the improvement of atmospheric forcing datasets and topographic processing algorithms to advance the field of continental-scale hydrology, and it highlights the importance of evaluating the continental-scale water balance as a whole for a process-based understanding of model performance and bias.

## 2 Methods

The PFCONUSv1 model was simulated using the coupled hydrology – land surface platform, ParFlow-CLM. In this section, we describe the governing equations for ParFlow-CLM formulated water balance, PFCONUSv1 configuration and inputs, datasets for model validation, and performance metrics.

### 2.1 Modeling the integrated water and energy balance with ParFlow-CLM

The full water balance for a given hydrologic unit can be generally expressed as $I_{in} - I_{out} = \Delta S$, where $I_{in}$ and $I_{out}$ represent the hydrologic inflows and outflows to some control volume, and $\Delta S$ is the change in water storage within the control volume. More specifically, the full water budget for a watershed under natural (nonanthropogenic) conditions can be written as





$$P_{rain} + P_{snow} + R_{in} - R_{out} + Q_{in} - Q_{out} - ET_{veg} - E_{dir} = \qquad (1)$$

$$\Delta S_{soil} + \Delta S_{surf} + \Delta S_{gw} + \Delta S_{snow}.$$

In (1), inflows to the watershed are precipitation in the form of rain or snow ($P_{rain}$, $P_{snow}$), surface runoff entering the basin from upstream areas ($R_{in}$), or subsurface influx ($Q_{in}$). Water may leave the watershed in the form of surface runoff ($R_{out}$), evapotranspiration from transpiration ($ET_{veg}$) or evaporation from bare surfaces ($E_{dir}$), or as groundwater flux to downstream basins ($Q_{out}$) or deeper reservoirs ($Q_{recharge}$). These fluxes have a net impact to yield increases or decreases to sources of basin water storage, such as soil and groundwater reservoirs ($\Delta S_{soil}$ and $\Delta S_{gw}$),

surface water ponding ($\Delta S_{surf}$), or storage as snow water equivalent ($\Delta S_{snow}$). This description of the water budget equation (1) is illustrated in Fig. 1a, and it may be amended to incorporate other components particular to a watershed; these could include anthropogenic fluxes and storage like irrigation, dam storage or pumping, or they could be unique traits of the basin such as fractured flow, lacustrine groundwater discharge, or seawater intrusion. Equation (1) may also be simplified by lumping precipitation, evapotranspiration and storage components, and also

by ignoring surface and subsurface inputs external to watershed divides which, for large enough control volumes, will be negligible (Fig. 1b). The water balance may then be simply expressed as,

$$P - ET - R = \Delta S \qquad (2)$$

for precipitation $P$, evapotranspiration $ET$, surface runoff $R$, and total change in all storage sources, $\Delta S$.

     In this study, the complete water balance (equation (1), Fig. 1a) is modeled using ParFlow-CLM (Kollet & Maxwell, 2006; Maxwell & Miller, 2005), an integrated groundwater-surface water model which uses the mixed form of Richards' equation to simulate three-dimensional variably saturated flow. The Richards equation is given as

$$S_s S(\psi_p) \frac{\delta \psi_p}{\delta t} + \phi \frac{\delta S(\psi_p)}{\delta t} = \Delta \cdot \left[ -K_s(x) k_r(\psi_p) \cdot \nabla(\psi_p - z) \right] + q_s \qquad (3)$$

     for specific storage $S_s$, relative permeability $S$, pressure head $\psi_p$, hydraulic conductivity tensor $K_s$, relative permeability $k_r$, at depth $z$ and time $t$. In (3), relative permeability varies with pressure head through time based on relationships established by van Genuchten (1980), and $q_s$ is a source-sink term. A free surface overland-flow

boundary condition for continuity of pressure and flux applies to the groundwater flux term across the land surface and subsurface interface. The kinematic wave approximation of the momentum equation is used to solve overland flow, which is a function of ponded depth given by Manning's equation,

$$v = \frac{\sqrt{S_0}}{n} \psi_p^{2/3} \qquad (4)$$




where $n$ is Manning roughness coefficient. Note that the friction slope $S_0$ in (4) is used to approximate the bed slope in the kinematic wave approximation.

ParFlow is coupled with the Common Land Model (CLM) (Dai et al., 2003), a land surface model which balances energy and calculates evapotranspiration at the land surface, in order to simulate the coupled water and energy budgets. CLM requires atmospheric conditions (precipitation, temperature, specific humidity, wind speed, and longwave and shortwave radiation) in order to provide hourly partitioning of net radiation into sensible, latent, and ground heat. Shown in equation (5), the CLM calculates direct evaporation ($ET_{ground}$ in equation (1)) using the gradient between specific humidity at the ground surface $q_g$ and at a reference height $q_a$, which is scaled by air density $\rho_a$, atmospheric resistance $r_d$, and a soil resistance term $\beta$.

$$ET_{dir} = -\beta \rho_a \frac{q_g - q_a}{r_d} \tag{5}$$

To calculate transpiration, CLM adjusts potential evapotranspiration $ET_{pot}$, by stomatal and aerodynamic resistance terms as follows:

$$ET_{pot} = \rho_a \frac{(L_{AI} + S_{AI})}{r_b} (q_f - q_c) \tag{6}$$

$$ET_{veg} = ET_{pot} \times \frac{L_d r_b}{L_{AI}} \left( \frac{L_{AI}}{r_b + r_s} \right) \tag{7}$$

Potential transpiration (6) is a function of leaf and stem area index $L_{AI}$ and $S_{AI}$, boundary layer resistance $r_b$, air density $\rho_a$ and the gradient of specific humidity between foliage and canopy, $q_f - q_c$, while actual transpiration (7) further depends on the fraction of dry canopy $L_d$ and the stomatal resistance $r_s$. Note that leaf and stem area index and stomatal resistance terms are parameterized by plant functional types, defined per cell without multilayer capability or fractional vegetation. For further explanation of $ET$ calculations in ParFlow-CLM, see Jefferson et al. (2017).

### 2.2 PFCONUSv1 configuration, parameters and inputs

The PFCONUSv1 model represents the first integrated groundwater-surface water model employed at the continental scale at hyper (1-km) resolution. A full description of the model configuration and inputs can be found in Maxwell et al. (2015) and Maxwell and Condon (2016), but a brief summary is given below.

Spanning roughly 6.3 million km² at 1 km lateral grid spacing, the PFCONUSv1 model encompasses the majority of eight major river basins in the United States at high resolution, including the Ohio, Missouri, Arkansas, Mississippi, and Colorado River Basins. The model is composed of 3442 cells in the $x$ (east-west) direction and 1888 cells in the $y$ direction (north-south). Its five vertical layers of variable thickness provide a cumulative vertical depth of 102 m. From the top, soil layers are 0.1, 0.3, 0.6, and 1 m, respectively. Topographic slopes were calculated using the Barnes et al. (2016) algorithm, applied to the shuttle elevation derivatives at multiple scales (HydroSHEDS) digital elevation model, to guarantee a connected drainage network. Vegetation classes for



characterization of plant functional parameters were provided by the IGBP land-cover classifications and USGS land-cover dataset. Distributed, heterogeneous soil parameters, including saturated hydraulic conductivity, porosity, and van Genuchten parameters, were assigned to spatial soil units described by the soil survey geographic database (SSURGO). Geologic units for the bottom, 100-m thick layer of the PFCONUSv1 model were developed from the Gleeson et al. (2011) national permeability map. Estimates from Gleeson et al. (2011) were adjusted using the e-folding relationship described in Fan et al. (2013), which accounts for topographic complexity, and variance in permeability was also reduced. No-flow boundary conditions were imposed at the bottom of the model domain (assuming impermeable bedrock) and on the sides. Note that with a model depth of just over 100 m, the model may more appropriately be considered a shallow aquifer storage model. Deeper $\Delta S$ contributions are not resolved, which may not represent deeper hydrologic flow paths of thick and expansive aquifers such as the Ogallala, the saturated thickness of which can exceed 300 m (McGuire et al., 1980); however, as Maxwell et al. (2015) explain, the current model thickness and vertical discretization is limited not by computational expense but by data availability, with a lack of detailed depth-to-bedrock and aquifer thickness estimates at meaningful resolution.

Initial conditions were provided by an intensive spinup process. First, a steady-state ParFlow groundwater configuration was run continuously without CLM; this model was forced by an average surface recharge flux derived from Maurer et al. (2002) and run continuously until the difference between outflow and recharge rates was less than 3 % of total water storage change. A full description of this steady-state model and its performance can be found in Maxwell et al. (2015). Second, and using the initial condition provided by the steady-state model, a transient system was simulated with the fully coupled ParFlow-CLM for water year 1985, the most climatologically average water year within the past 30 years. As described in Maxwell and Condon (2016), atmospheric forcing was bilinearly interpolated from the North American Land Data Assimilation System Phase 2 (NLDAS 2) (Cosgrove, 2003; Xia, Mitchell, Ek, Sheffield, et al., 2012). For spinup purposes, the transient simulation was run continuously for four years of repeated 1985 atmospheric forcing to provide an initial condition for the simulation in this study. Thus, the initial condition provided here represents pressure head, soil moisture and surface energy balance conditions that would be present during the most climatologically average water year in recent history. Since the model does not incorporate anthropogenic abstractions in the form of pumping, injections, irrigation or surface water diversions and dam storage, the initial conditions provided also represent a pre-development scenario.

For this study, PFCONUSv1 was run for modern-day water years using initial conditions provided by the transient spinup process described above. The simulation here was run at hourly temporal resolution for water years 2003 through 2006 Atmospheric forcing originated from the 12 km NLDAS-2 product (Xia et al., 2012); however, finer resolution products were blended in where available and elevation effects were incorporated to produce higher resolution, more physically realistic meteorological variables. Such products included the 4 km Stave IV and Stage II radar and gauge products and Level 2 shortwave radiation from the GOES Surface and Insolation Products (GSIP). These adjustments to the 12 km NLDAS data and the finer resolution products are described, for example, in Pan et al. (2016) and include the following: gap-filling and daily rescaling procedure to ensure the Stage IV hourly data match daily totals from NLDAS-2; adjustments to timing for the GSIP Level 2 data based on solar angles; and elevation-dependent downscaling of 12 km NLDAS-2 products, such as hydrostatic effects for



atmospheric pressure and lapse rates for specific humidity, air temperature and longwave radiation. The final
atmospheric variables were interpolated using bilinear interpolation to the 1-km PFCONUSv1 grid.

An important consideration when attempting high resolution integrated models of this kind is of course the computational demand. ParFlow-CLM solves the globally implicit solutions to nonlinear and coupled equations in (3) through (7) with a Newton-Krylov parallel solver (Jones & Woodward, 2001); the associated significant computational challenge is tackled with a multigrid preconditioner and highly scaled parallel efficiency (Kollet et

al., 2010; Reed M. Maxwell, 2013; Osei-Kuffuor et al., 2014). The simulations presented here were run on 3456 processors, distributed to 72 and 48 units in the $x$ and $y$ directions, respectively, on the Cheyenne high performance computing system managed by the National Center for Atmospheric Research (NCAR) Computational & Information Systems Lab. Required core hours for a single water year averaged over 300 thousand core-hours for this processor topology; however, the scaled parallel efficiency even at this decomposition is over 60 percent. The

hourly outputs generated over 11 terabytes of information per water year, while the required storage for the interpolated atmospheric forcing alone was over 3 terabytes per water year.

### 2.3    Datasets for comparison

Simulated runoff, evapotranspiration and sources of storage change from the PFCONUSv1 model were
compared against available point-scale measurements and coarse resolution remote sensing products in order to establish confidence in modeled results and to identify major sources of model bias. Table 1 provides a summary of all data products compared to PFCONUSv1 outputs. It is important to note here that while we use absolute error metrics common to calibrated models developed specifically for prediction, calibration of the PFCONUSv1 model is not a goal of this study, nor is it feasible given the computational demands posed by such a highly parallelized

platform. Rather, the intent is to evaluate the model's ability to demonstrate realistic behavior, to identify regions, times, and sources of uncertainty, and to prioritize areas of improvement for future model development.

### 2.3.1    Surface water runoff, *R*

Modeled surface water runoff ($R$ in equation (2)) was compared to daily observations at 2,392 U.S.
Geological Survey (USGS) stream gauges containing observations over the simulation period (October 1, 2002, through September 30, 2006) within the PFCONUSv1 domain (Table 1) (obtained from https://waterdata.usgs.gov/nwis/sw, last accessed February 2, 2020). As discussed in the supplemental information for Maxwell and Condon (2016), the algorithm used for topographic processing resulted in spatial inconsistencies between the real and modeled stream network. USGS gauges were therefore mapped to the PFCONUSv1 grid using

a combination of nearest neighbor mapping and manual adjustments to ensure that all gauges lay on an appropriate ParFlow stream cell; for instance, a gauge comparison point that was incorrectly mapped upstream of a confluence may be moved to an appropriate location downstream. The large majority of mapped gauges were within 3 km of their 'actual' location. As Maxwell and Condon (2016) explain, approximately 10 percent of USGS gauges required more significant manual adjustments because of considerable discrepancies between the true stream network and

that constructed for the model.





### 2.3.2   Evapotranspiration, *ET*

For evapotranspiration (*ET* in equation (2)), three datasets are used to evaluate PFCONUSv1 results (Table 1). Observations from FLUXNET, an international network of meteorological towers that rely on the eddy covariance method to estimate evapotranspiration, were used to evaluate the temporal performance in ET at 30

locations containing data during the simulation period. FLUXNET data were obtained from the FLUXNET 2015 online data portal (https://fluxnet.fluxdata.org/, accessed February 6, 2020). PFCONUSv1 ET estimates were also compared to MODIS evapotranspiration MOD16A2 monthly product provided by the University of Montana Numerical Terradynamic Simulation Group (NTSG) lab (http://files.ntsg.umt.edu/data/NTSG_Products/MOD16/, last accessed March 20, 2020). The MODIS product, a NASA and EOS initiative to estimate global terrestrial

evapotranspiration using satellite remote sensing data, uses a Penman-Monteith based approach, stomatal resistance and vegetation information to estimate evapotranspiration at an 8-day interval at 1-km resolution (Mu et al., 2007, 2011). MOD16A2 improves upon the original MOD16 ET algorithm by considering surface energy partitioning and atmospheric demand as well as land cover, leaf area index, and meteorological reanalysis products provided by NASA's Global Modeling and Assimilation Office (GMAO). Given the 8-day interval limitation and point-based

uncertainties in ET of up to 40-60% (Velpuri et al., 2013; Westerhoff, 2015), the monthly MOD16A2 product was spatially aggregated to HUC8 watersheds across the PFCONUSv1 domain with equal area weighting. We also compare HUC8-aggregated monthly PFCONUSv1 evapotranspiration with estimates from the Operational Simplified Surface Energy Balance (SSEBop) algorithm (Senay et al., 2013). The SSEBop model is a relatively simple model, using 1km 8-day MODIS remotely sensed thermal imagery (land surface temperature and emissivity),

combined with thermal index reference ET Senay et al., 2013). Velpuri et al. (2013) evaluated MOD16A2 and SSEBop performance across the contiguous United States at point and basin scales, finding that SSEBop outperformed MOD16A2 in western, arid basins. Note that for FLUXNET observations, ET (mm day$^{-1}$) was derived from latent heat (W m$^{-2}$) by scaling by the latent heat of vaporization $\lambda$ (2.45 MJ kg$^{-1}$) with the proportional relationship $ET = \frac{LE}{\lambda}$.


### 2.3.3   Storage, *S*

To evaluate PFCONUSv1 storage change ($\Delta S$ in equation (2)), four products are used to compare to individual storage components, including total water storage, snow water storage, and soil water storage. Modeled snow water equivalent was compared to Snow Telemetry (SNOTEL) station data, a network maintained by the

Natural Resources Conservation Service (NRCS). SNOTEL data were accessed from the NRCS online report generator 2.0 (http://wcc.sc.egov.usda.gov/reportGenerator/, last accessed February 28, 2020). Of the available SNOTEL stations, 556 are within the PFCONUSv1 domain and have observations during the simulation period. These SNOTEL locations were compared to simulated snow water equivalent at their nearest neighbor PFCONUSv1 grid cells. For soil water storage, soil moisture anomalies were derived from the active passive

satellite products from the ESA Programme on Global Monitoring of Essential Climate Variables (ECV) Soil Moisture Climate Change Initiative (CCI) project v04.5 (Gruber et al., 2019; https://www.esa-soilmoisture-





cci.org/node/237, last accessed February 20, 2020). This remote sensing product uses a combined estimate of soil moisture from 4 active and 7 passive microwave sensors, providing soil storage at 0.25° resolution.

PFCONUSv1 total water storage anomalies (an aggregate of all subsurface, snow water and surface water storage components) was also compared to terrestrial water storage anomalies provided from remote sensing products from the Gravity Recovery and Climate Experiment (GRACE). The GRACE products are derived from slight fluctuations in Earth's gravity caused by changes in mass and measured by twin satellites launched in 2002; these gravity field changes over land may be attributable to terrestrial water storage change. GRACE solutions are provided by three processing centers: the NASA Jet Propulsion Laboratory (JPL), the GeoforschungsZentrum

Potsdam (GFZ), and the Center for Space Research at University of Texas, Austin (CSR). In this study, PFCONUSv1 total water storage changes were compared to the Release-06 gravity field solutions (RL06) at 1°, calculated using the spherical harmonic approach (Landerer & Swenson, 2012) with varying degrees and orders, spherical harmonic coefficients and filtering processes. We also compare PFCONUSv1 to the mass concentration block (mascon) solutions provided by JPL at 0.5° and CSR at 0.25° (Save et al., 2016; Wiese et al., 2016), which

eliminate much of the need for empirical post-processing and filtering required in the spherical harmonic solutions. The GRACE products listed above are hereafter referred to as JPL, GFZ, and CSR for the RL06 spherical harmonic solutions, and JPLm and CSRm for the mascon solutions. For both the ESACCI soil moisture product and the GRACE total water storage anomalies, PFCONUSv1 estimates are aggregated to the coarse resolution product by area weighted mean prior to comparisons.

Finally, PFCONUSv1 calculated depth to water table are compared with water levels from 41,269 USGS groundwater wells across the continental United States; like streamflow, these data are freely available for download from the USGS National Water Information System (https://waterdata.usgs.gov/nwis/gw, last accessed March 23, 2020). Of these wells, locations with more than 10 observations during the simulation timeframe and that met requirements for appropriate aquifer comparison (such as well depth, aquifer type, and anthropogenic influence)

were used to calculate correlations with PFCONUSv1 timeseries; 2,486 wells fit these criteria (see Table 1) and will be discussed further in Sect. 3. Note that in this study, we focus on the change in water storage over a given period of time, rather than the total amount of water currently stored. Storage anomalies are presented as deviations through time from mean storage states; we also discuss the water storage amplitude, or peak-to-peak intra-annual storage change, for a given region, as a proxy for seasonality. In the majority of the PFCONUSv1 domain, over this

relatively brief simulation period, the variance in the intra-annual (seasonal) signal explains the majority of the variance in storage anomaly timeseries.

### 2.3.4 Atmospheric forcing

   One important source of bias is that of atmospheric forcing; to evaluate the impact of meteorological

performance on simulated water balance variables, we compare the interpolated NLDAS product to observed daily precipitation ($N$=9,193) and observed averaged daily temperature ($N$=1,678) at meteorological stations maintained by the Global Historical Climatology Network (GHCND) (Table 3.1) (data accessed via Climate Data Online portal,





https://www.ncdc.noaa.gov/cdo-web/search, last accessed February 11, 2020). Atmospheric forcing variables were also compared to observed data at SNOTEL and FLUXNET sites.


### 2.4 Performance metrics

Performance metrics for evaluating PFCONUSv1 include percent annual bias or total annual bias, Spearman rank correlation coefficient, and the ratio of Root Mean Squared Error to the standard deviation of observations (RSR). While these were not calculated for all validation datasets, and the temporal resolution at which they were evaluated differed between datasets (e.g., daily, weekly, or monthly), they are each used at some point in our analysis, so we define them here.

As a measure of average magnitude accuracy with an optimal value of 0, percent bias is given by

$$PBIAS = \frac{\sum_{i=1}^{n} S_i - O_i}{\sum_{i=1}^{n} O_i} * 100 \qquad (8)$$


where $S_i$ and $O_i$ are simulated and observed values. Percent bias in PFCONUSv1 outputs was calculated using daily observations in equation (8), such that days during which observations were unavailable were excluded for both simulated and observed annual totals. Percent bias is an effective metric for evaluating long-term mean values, but it cannot be used to evaluate timing or shorter temporal events; further, if the model under- and over-predicts with similar magnitudes, PBIAS can be deceivingly low.


For these reasons, we also calculate for each stream gauge Spearman's rank correlation coefficient, or Spearman's $\rho$, given by (9):

$$\rho = 1 - \frac{6 \sum_{i=1}^{n} d_i^2}{n(n^2 - 1)} \qquad (9)$$


Unlike the coefficient of determination $R^2$, which describes the degree of collinearity between the data, Spearman's $\rho$ independently ranks the simulated and observed values, with $d_i$ in (9) being the difference in ranks for a given value $i$, and $n$ is the number of values in the series. Unlike other metrics describing temporal correlation, such as $R^2$ or Nash-Sutcliffe Efficiency, $\rho$ is less restrictive; it does not assume linearity and instead and tests for monotonic correlation. The optimal value for $\rho$ is 1, and the cutoff for good performance is likely analogous to that of $R^2$, which varies in the literature but is generally around 0.6.


A final performance metric, the RMSE-observations standard deviation ratio (RSR) is also provided. RSR is given by equation (10), RSR describes root mean squared error (RMSE) relative to the standard deviation of the observations.

$$RSR = \frac{RMSE}{St.Dev.Obs.} = \frac{\sqrt{\sum_{i=1}^{n}(O_i - S_i)^2}}{\sqrt{\sum_{i=1}^{n}(O_i - \bar{S})^2}} \qquad (10)$$



In (10), $\bar{S}$ is the mean of observations. While RSR is less widely used than PBIAS and $\rho$, its benefit lies in its normalization of common error index statistic RMSE; the ratio describes error relative to natural variability in the true system, such that an RSR of 1 suggests that the mean daily error is equal to one standard deviation of observed values and thus comparable to what we may expect from noise or intra-annual variability. An RSR value of 0 is optimal, while values under 0.5 (RMSE is less than half of the standard deviation of observations) is considered to be excellent (Moriasi et al., 1983).

### 3      Results

By providing detailed partitioning of the water and energy budgets at high spatial and temporal resolution and at continental spatial extent, the PFCONUSv1 ParFlow-CLM model offers an unprecedented opportunity to study large-scale nonlinear relationships and to provide hydrologic process estimates at locations remote from observation networks. The 2003-2006 water year simulations in this study estimate hourly pressure head and saturation at each of the approximately 31.5 million 1-km three-dimensional model cells; the simulations also provide evapotranspiration, and energy balance estimates at each of the 6.3 million land surface grids cells. Figure 2 shows the PFCONUSv1 model extent, mean annual precipitation from interpolated atmospheric forcing, and mean annual simulated components of equation (2). Below, these water balance components, their performance, and their relative bias sources are discussed in detail. Note that different performance metrics were discussed for model components based on their temporal and spatial coverage, continuity, resolution, and uncertainty. For instance, the shear amount of temporal and spatial coverage provided by the USGS stream gauge network allowed for several different error metrics to evaluate long-term behavior, hydrograph shape, and flashiness. Comparisons of model results with remote sensing products and well observations were more limited by higher uncertainty and lower temporal and spatial resolution and continuity, but they were still valuable in identifying regions for model improvement and analyzing error propagation between water balance components.

### 3.1      Runoff, *R*

The ability to accurately simulate overland flow at the major basin or continental scale and above has for several years been a topic of much interest in the hydrologic community. Continental or global streamflow estimates could be coupled to general circulation models to provide predictions of surface water resource vulnerability to climate change (e.g., Koirala et al., 2014); large-scale runoff models could additionally provide flood forecasts to regions lacking in developed surface water monitoring networks (Kauffeldt et al., 2016). While the integrated groundwater-surface water modeling is computationally demanding, results from PFCONUSv1 represent a rare opportunity to evaluate streamflow performance, 1) because the integrated system platform resolves shallow aquifer, vadose zone and surface water transfer, and 2) streams form naturally as surface water is routed by topography, without requiring pre-defined stream reaches.

PFCONUSv1 streamflow *R* was evaluated against 2,392 USGS stream gauges which are well-distributed across the United States. We analyze model performance using percent bias, Spearman rank correlation, and RSR. However, PBIAS can be sensitive to precipitation provided by the interpolated NLDAS atmospheric forcing



product. Since $P$ is an input to the PFCONUSv1 rather than a model result, runoff ratio ($RR = \frac{R}{P}$) was also

calculated to extract model performance independent of precipitation bias. RR measures the amount of precipitation
partitioned to runoff, with lower RR values generally indicating a greater portion of precipitation lost to infiltration
or evapotranspiration. "True" runoff ratios were estimated by first identifying all GHCND precipitation gauges
upstream of a USGS stream gauge. The mean annual precipitation was then calculated and applied over the drainage
area defined by Geospatial Attributes of Gages for Evaluating Streamflow (GAGES-II) dataset (Falcone, 2002). RR

is equal to the ratio of total USGS gauge flow to GHCND precipitation. A similar process was done for simulated
RR, using NLDAS-interpolated precipitation, simulated flow at the gauge cell, and model drainage area from the
input digital elevation model derived from HydroSHEDS. Note that while the interpolated NLDAS precipitation,
unlike the GHCND gauge network, is continuous in space, only modeled cells which matched nearest-neighbor
GHCND gauge network were used to estimate upstream precipitation, in order to create as controlled a comparison

as possible. Runoff ratios were not calculated for USGS stream gauges with fewer than three upstream GHCND
precipitation gauges.

Observed total annual flow during the simulation period is shown in Fig. 3a; annual streamflow varies by
several orders of magnitude across U.S. major basins, with higher flows in the east and the Pacific Northwest, and
lowest flows in the Great Plains. Runoff ratios are generally highest in the East; the majority of the arid West exhibit

RR of less than 0.1, with the exception of topographically complex regions and headwater watersheds of the Rocky
Mountains (Fig. 3c).

PFCONUSv1 appropriately reproduces point-scale annual flows across the United States, with a median
annual PBIAS of 7.7 %, and with 25th and 75th percentiles of -26.2% and 77.4%, respectively (Fig. 3b). Shown in
Fig. 3.3e and f, the 25th, 50th, and 75th percentiles for Spearman's $\rho$ are 0.42, 0.65, and 0.76, while the same for RSR

are 0.86, 1.2, and 2.5. The median PFCONUSv1 minus USGS difference in RR is 0.016 (Fig. 3d), which
corresponds to a mean percent bias in runoff ratio of 8.3%. The PFCONUSv1 model performs exceptionally well
(RSR<0.6, $\rho$>0.7, and |PBIAS|<20%) at 54 gauges (approximately 2% of available sites); an additional 97 locations
exhibit very good performance (4% of gauges, RSR<0.7, $\rho$>0.65, and |PBIAS|<30%); an additional 382 locations
show good performance (15.7% of gauges, RSR<1, $\rho$>0.6, and |PBIAS|<50%); and an additional 268 gauges show

overall acceptable performance (11% RSR<1.2, $\rho$>0.5, and |PBIAS|<75%). A total of 801 gauges perform
acceptably for all performance criteria (34% of all gauges). However, the large majority of gauges perform well for
one metric but not others: 2099 gauges show either RSR<1.2, |PBIAS|<75%, or $\rho$>0.5 (86% of gauges), leaving
14% of the USGS streamflow gauges showing poor PFCONUSv1 performance based on all metrics.

Streamflow performance varies widely across major basins. For instance, median PBIAS, $\rho$, and RSR for

the Ohio River Basin are -7.8%, 0.79, and 0.84, respectively, and median simulated RR values are within 6% the
median estimate RR=0.42 from observations. Simulated flows in the Tennessee River Basin also appropriately
simulate observed flows: mean PBIAS, $\rho$ and RSR are -11.9%, 0.69, and 0.89, respectively; 60% of the gauges in
the basin perform acceptably based on all performance metrics (RSR<1.2, $\rho$>0.5, and |PBIAS|<75%); and observed
and simulated mean RR are 0.49 and 0.53 respectively, for a percent bias in RR of 9%. Conversely, the majority of

the upper Missouri River Basin shows weak timing performance (median $\rho$ of 0.49) and higher overall bias: the





median PBIAS for Missouri is 65% and median RSR is 2.2, indicating that the majority of Missouri gauges exhibit daily RMSE that is twice the volume of expected daily variability. The Great Plains region is certainly the region with worst streamflow performance: PFCONUSv1 percent bias in the majority of these gauges is greater than 300%, and in some cases, simulated flow is greater than 10 times the volume of observed. While the mean difference in

runoff ratio in this region is only 0.04, this is on average 4 times larger than RR estimated from observations. Results in Fig. 3 therefore suggest that in the arid Great Plains region, a very small change in runoff ratio can result in dramatic error in streamflow bias, and the PFCONUSv1 struggles to capture low flows in this region.

   Note that no filtering was done for these metrics in order to eliminate gauges with incorrect drainage area from topographic processing discrepancies, nor have we removed sites proximate to dams, influenced by nearby

pumping or irrigation or affected by bias in atmospheric forcing. As an example of PFCONUSv1 performance in ideal conditions, we show in Fig. 4 selected examples of individual gauge comparisons for each major basin in the PFCONUSv1 domain. Gauges chosen for Fig. 4 were those that tended to be minimally impacted by bias from anthropogenic effects or by errors in basin delineation by topographic processing. Such gauge attributes were determined based on geospatial stream properties obtained from the Geospatial Attributes of Gages for Evaluating

Streamflow (Gages-II) dataset (Falcone, 2002), as well as the National Hydrography Dataset (see the supplemental information in Maxwell & Condon (2016) for detailed description of geospatial stream gauge attributes). Streamflow timeseries examples in Fig. 4 include gauges with the following properties: 1) represented greater than 300 km$^2$ upstream drainage area, 2) PFCONUSv1 drainage area differed from actual drainage area by less than 20 %, 3) total dam storage was less than 3% of total annual flow for the closest upstream dam, 4) total withdrawals for

previous five years were less than 3% of total annual flow, 5) total irrigated area in 2002 constituted no more than 15% of the total drainage area, and 6) upstream area Spearman's $\rho$ for precipitation performance must be greater than 0.5. The examples in Fig. 4 therefore represent naturalized gauges, those with minimal bias in a priori inputs, low anthropogenic impact, and good performance potential.

## 3.2 Evapotranspiration, *ET*

   Evapotranspiration is a major component of the water balance, accounting for roughly 60 percent partitioning of land precipitation into the atmosphere annually (Oki & Kanae, 2006); however, it is also widely considered to be an incredibly difficult value to constrain (Gabriel B. Senay et al., 2013; Velpuri et al., 2013; Xu & Singh, 2005) and is often estimated simply as the residual of other components of the water balance. Unlike

streamflow and precipitation, direct point measurement methodologies are limited, costly, and difficult to maintain. Direct estimates can be inferred from sap flux measurements; lysimeters, which weigh plant and soil mass to track temporal fluctuations in water storage; or chemical tracers, such as deuterium (Wilson et al., 2001). Currently, the method which likely provides the most defensible direct measurements of ET is the eddy flux or eddy correlation method. Eddy flux towers relate observed turbulent heat fluxes at the surface (latent and sensible heat) to the

covariance between instantaneous fluctuations of vertical wind speed, humidity and temperature (Baldocchi, 2003; Reynolds, 1895; Swinbank & Swinbank, 1951). The PFCONUSv1 simulated daily ET was compared to observations from 30 eddy covariance towers managed by the FLUXNET mission. Locations of these sites and their





relative performance are shown in Fig. 5, along with timeseries examples from three FLUXNET with complete observations during the entire observation period.

Results in Fig. 5 demonstrate the ability of the PFCONUSv1 model to appropriately simulate daily and seasonal ET across difference climatic zones. The mean 25th, 50th, and 75th percentiles for PFCONUSv1-simulated daily ET PBIAS are 3%, 26%, and 55%, respectively. Given that remote sensing estimates regularly exhibit uncertainty of 50-60% for point-scale ET estimates, or >20% uncertainty in ET at the basin-scale (Velpuri et al., 2013), PFCONUSv1 ET results are promising, especially for an uncalibrated model. For daily timeseries, 25th, 50th,

and 75th percentiles are 0.6, 0.72, and 0.81 for $\rho$, and 0.69, 0.92, and 1.33 for RSR, respectively. Because the metric is an indicator of monotonic agreement, the high overall Spearman's $\rho$ values are particularly telling, because $\rho$ is sensitive not only to seasonal trends which dominate the timeseries variance but also the influential day-to-day (sub-seasonal) noise. Out of 30 FLUXNET sites with observations during the simulation time period, the PFCONUSv1 model performs acceptably for all performance criteria at 19 locations (63% of locations); at 29 out of 30 sites, the

PFCONUSv1 model performs well based on at least one metric.

While Fig. 5 builds confidence that the PFCONUSv1 model can appropriately simulate evapotranspiration over a range of geophysical, vegetative and climatic categories, the spatial discontinuity of FLUXNET certainly limits ET performance evaluation across the remaining PFCONUSv1 model domain. Eddy covariance ET estimates are applicable within the fetch of the prevailing winds, which is generally on the order of ~1 km radius surrounding

towers (Wilson et al., 2001), and statistical interpolation is generally not recommended without considerable parameterization of atmospheric and vegetative conditions to inform upscaling (Jung et al., 2009).

To build confidence at larger spatial scales, the PFCONUSv1 model has also been compared to the MOD16A2 and SSEBop algorithms for MODIS thermal imagery processing. These data, along with PFCONUSv1, have been aggregated to HUC8 spatial scale and monthly temporal resolution to help reduce uncertainty associated

with cloud cover in the 8-day product. Cumulative annual evapotranspiration for MOD16A2, SSEBop, and PFCONUSv1 are shown in Fig. 6a-c. Both MOD16A2 and SSEBop algorithms should be considered evapotranspiration modeling techniques produced from remote sensing observations, rather than observations themselves. However, regions where PFCONUSv1 comparisons to the MOD16A2 and SSEBop agree provides confidence in the model's bias or timing of ET estimates. We have therefore used PBIAS, $\rho$, and RSR error metrics,

with PFCONUSv1 monthly ET observations as simulated and MODIS datasets as observed values. Multiple studies to date have compared MOD16A2 and SSEBop performance over a range of geophysical characteristics, vegetative types and aridity indices by comparing to Penman-Monteith -based estimates (Knipper et al., 2016), lysimetric observations (Senay et al., 2014), FLUXNET observations or upscaled information from FLUXNET sites and vegetative indices (Senay et al., 2013; Velpuri et al., 2013), with results showing good general agreement and within

~50% error for annual ET totals at point observations. Despite its considerably more simplistic approach for estimating ET from MODIS thermal land imaging, SSEBop performs nearly as well or, in the case of the Western U.S., better than MOD16A2 (Velpuri et al., 2013). Therefore, we also show MOD16A2 performance relative to SSEBop for reference (Fig. 6 c, f, i, l, o), where $O_i$ observations are MOD16A2 and $S_i$ observations are MOD16A2 monthly ET (equations 8, 10).

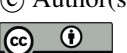



PFCONUSv1 shows good general agreement with MOD16A2 and SSEBop algorithms in annual ET, with differences within ±30mm across the domain. All products provide similar spatial signatures of ET, with overall higher ET in the south west and lowest ET in the Great Basin and Colorado River Basins. PFCONUSv1 estimates tend to agree more with SSEBop with regards to timing and residual variation, and they are more similar to MOD16A2 with regards to PBIAS (particularly in the Western United States) (Fig. 6). The 50th percentile for

PFCONUSv1 PBIAS against MOD16A2 and SSEBop are 7.5% and 8%, respectively; 25th and 75th percentiles of PBIAS are -4.4% and 24% for MOD16A2, and -4.4% and 35% for SSEBop. In several regions, PFCONUSv1 shows similar comparisons with both MODIS products. For instance, in the Upper Mississippi, both products suggest that PFCONUSv1 overpredicts ET in the north and underpredicts in the South; and both products suggest PFCONUSv1 underpredicts ET in the Rocky Mountain headwaters and across most of the Ohio River Basin (Fig. 6 d, e). The

approximately 30% underestimation of ET in the CO headwaters further agrees with the PFCONUSv1 performance relative to FLUXNET observations at the Niwot Ridge site in Colorado. However, most of the Missouri and the the Arkansas-Red-White basins show opposite behavior between PFCONUSv1-MOD16A2 and PFCONUSv1-SSEBop comparisons; in these regions, we can be less certain of model bias as described by remote sensing of ET. Broadly, across the Western U.S., PFCONUSv1 shows better agreement with MOD16A2 with regards to ET magnitude

(PBIAS) because SSEBop estimates negligible ET in the Basin and Range region (Fig. 6 e); however, PFCONUSv1 shows dramatically better performance relative to SSEBop in terms of Spearman's $\rho$ and RSR (Fig. 6 g,h). The 25th, 50th, and 75th percentiles of $\rho$ are 0.38 (unacceptable), 0.85 (excellent), and 0.92 for monthly PFCONUSv1 compared to MOD16A2; the quantiles for PFCONUSv1 compared to SSEBop are 0.85 (excellent), 0.91, and 0.93. Similarly, 25th, 50th, and 75th percentiles of RSR are 0.41, 0.85, and 2.2 for performance relative to MOD16A2, and

0.38, 0.47, and 0.62 for performance relative to SSEBop.

Despite differences between PFCONUSv1 comparisons to the two MODIS algorithms, results shown in Fig. 3.6 build confidence that PFCONUSv1 appropriately estimates the magnitude and temporal progression of ET. However, it is important to again note that MODIS ET estimates are themselves models, and as such they are susceptible to epistemic errors in input data (e.g., inaccuracies in LAI or other parameterizations), measurement and

remote sensing errors (e.g., cloud cover), and other uncertainties. Nevertheless, MODIS represents the ET dataset with greatest temporal and spatial coverage and resolution. There is evidence that PFCONUSv1 could outperform other land surface models in its ability to estimate total ET across the U.S.; in a study comparing LSM-based recharge estimates in the western United States, Niraula et al. (2017) showed that LSMs Mosaic, VIC, and Noah simulated spatially distributed ET with 0.87, 0.77, and 0.75 Pearson's correlation relative to MODIS. Pearson's

correlations between PFCONUSv1 and MOD16A2, and between PFCONUSv1 and SSEBop, are 0.9 and 0.95, respectively, which motivates future comparisons of PFCONUSv1 performance relative to other LSMs.

### 3.3    Storage, *S*

Terrestrial water storage represents all components of the water balance stored on and below the Earth's

surface. As such, total storage *S* is an aggregate of water stored on land in surface water bodies or in the canopy, as well as snow water equivalent, soil moisture in the vadose zone, and shallow or deep saturated aquifer storage.



Estimates of overall $S$ could simply involve measuring and combining individual components, but these calculations 1) require highly developed monitoring networks and an impressive amount of in situ observations, and 2) can have large margins of error if not all of the assorted hydrological stores are accurately resolved (Troch et al., 2007). The most common method for $S$ estimation calculates storage as a remainder of other water balance terms, $P$, $ET$, and $R$ in equation (2), (e.g., Rodell et al., 2011; Tang et al., 2010).


Recent advances in remote sensing have granted hydrologists an estimate of changes to total $S$, without partitioning out storage sources, by measuring fluctuations of Earth's gravity fields as a proxy for mass change. Since water represents the greatest fluctuation of terrestrial mass, gravity anomalies can be translated to variability in $S$. The newly available GRACE twin satellite mission provides approximately monthly values of total water storage at the global scale and at coarse ($>10^5$ km) resolution (e.g., Strassberg et al., 2009). Storage anomaly estimates are based on K-band microwave measurements of the distance between the two low-flying satellites, which varies as a function of gravity field fluctuations (as well as atmospheric, oceanic, and solid Earth tides, which must be corrected to resolve the global water budget (Dahle et al., 2019)).


The PFCONUSv1 total water storage anomalies (calculated as a sum of all simulated surface and subsurface hydrologic stores) was compared to five monthly gravity field solutions: the RL06 spherical harmonic solutions provided by JPL, GFZ, and CSR, as well as mascon solutions JPLm and CSRm. Figure 6 shows seasonal storage amplitude in space as well as basin-aggregate storage change through time, comparing PFCONUSv1 and GRACE solutions for six major river basins. Some basins have been left out due to incompleteness in the model domain, or due to size: The basis function for GRACE solutions is generally on the order of 300,000 km$^2$, such that storage anomaly estimates for smaller basins (e.g., the Tennessee River Basin) are not well resolved.



Seasonal storage amplitude represents the average peak-to-peak storage gain or loss over the course of a water year, and it is therefore a depiction of seasonality or intra-annual $S$ signal. The GRACE solution shown in Fig. 7 is the JPL mascon solution provided at 0.5° resolution, and amplitude for other GRACE products show similar spatial signals; however, note that mascon solutions are calculated given a 3° equal-area basis function and subsequently downscaled using forward modeling techniques to account for leakage errors (Wiese et al., 2016). GRACE mascons are not independent of each other, and uncertainty increases dramatically with decreasing basin size. However, qualitative comparisons between GRACE and PFCONUSv1 amplitude indicate several regions of agreement for high or low seasonality. Topographic highs in the Rocky Mountains, where the snow water equivalent signal likely dominates overall storage variance and is entirely seasonally dependent, show high amplitude for both PFCONUSv1 and GRACE (Fig. 7 a,b). The Upper and Lower Colorado River basins, in particular, show very similar spatial patterns for overall amplitude. Another area of agreement is the comparably high amplitude in the lower Mississippi River Basin. In both GRACE and PFCONUSv1, the Arkansas-Red-White region sees higher seasonality of total water storage in the east, and lower in the west; and the locations of highest amplitude, both for GRACE and our model, lie in the Pacific Northwest region. However, broadly speaking the PFCONUSv1 amplitude is lower than GRACE for the majority of the domain and particularly in the East. Other continental- or global-scale land surface models have also underpredicted seasonal storage amplitude across global river basins relative to GRACE; for example, the WaterGAP (Water Global Assessment and Prognosis) hydrologic model consistently






under-predicted amplitude for most of the global land area (Döll et al., 2014), and a validation of four LSMs and
global hydrologic models found that the numerical models reproduced GRACE storage signals only to a very
limited degree (Zhang et al., 2017). However, LSM tendency for GRACE mismatch is likely attributed to
insufficient groundwater representation, which is not as likely to be the cause for PFCONUSv1 and GRACE
disparities.

    Temporal progression of storage was calculated with area-weighted mean of the Colorado, Arkansas, Ohio,
Missouri, and Upper Mississippi River Basins (Fig. 7, c-h). Uncertainty (shaded regions) shown indicates the
leakage error associated with downscaling 3° basis functions to 0.5° solutions for the JPL mascon product. We show
only the JPLm uncertainty, simply because uncertainty estimates for the RL06 products are not yet available. The
CSR mascon product is suggested to have an error of approximately 2cm that is more or less constant through time
and space.

The PFCONUSv1 model shows good agreement in the seasonal shape of storage anomalies for most
basins, with Spearman's $\rho$ rank correlation ranging from 0.43 to 0.94 relative to the mean of all GRACE solutions:
Individual $\rho$ values for major basins are 0.43 (Missouri), 0.63 (Upper Colorado), 0.76 (Pacific Northwest), 0.79
(Great Basin), 0.81 (Lower Colorado), 0.86 (Upper Mississippi), 0.88 (Ohio), and 0.93 (Arkansas-Red-White).
However, correlation is not necessarily the best predictor of adequate model performance; for instance, the Upper
Mississippi has the third highest $\rho$ value out of six major basins, but more than 80% of the total anomaly timeseries
lies within the uncertainty bars provided for the JPLm product. Further, several discrepancies exist between
PFCONUSv1 and GRACE trends and amplitude. For example, despite its monotonic agreement with GRACE
storage amplitude for the Ohio River Basin, the PFCONUSv1 model simulates a seasonal storage amplitude that is,
on average, more than 30% lower than what GRACE observes. The Upper Colorado River Basin captures seasonal
timing, but the overall storage gain over the simulation period is roughly three times that of what GRACE observes.

    Differences between the PFCONUSv1 and GRACE storage water anomaly estimates can come from
various sources: 1) *model error and uncertainty* in PFCONUSv1 model parameters and configuration, error and
uncertainty associated with GRACE measurement error, or error associated with the intensive post-processing and
filtering on the raw spherical harmonic GRACE solutions, 2) *hydrologic stores unaccounted for* in the PFCONUSv1
model, such as deep (>100m) aquifer storage, or 3) *anthropogenic impacts*, particularly from groundwater
withdrawals from municipal and agricultural aquifer depletions (Chen et al., 2016).

**3.4     Storage partitioning:** $S_{gw}$, $S_{soil}$, and $S_{snow}$

    Total water storage anomalies were also validated based on their partitioned components: $\Delta S_{gw}$, $\Delta S_{soil}$, and
$\Delta S_{snow}$. First, PFCONUSv1 water table depth (WTD) was compared to USGS well observations across the United
States in Sect. 3.4.1. As discussed below, WTD does not necessarily translate to $\Delta S_{gw}$, but it is still a very
informative hydrologic state. PFCONUSv1 soil moisture was compared to a combined active passive remote sensing
product in Sect. 3.4.2, and PFCONUSv1 snow water equivalent was compared to snow telemetry measurements in
Sect. 3.4.3.






### 3.4.1 Water table depth

Figure 8 shows observed WTD across the model domain, as well as difference in observed and modeled heads and correlation for available locations. As a caveat to the results shown in Fig. 8, while WTD is a visibly appealing metric for modeled groundwater performance, it alone is not translatable to total storage $S_{gw}$ or for storage

change $\Delta S_{gw}$, for several reasons. First, without information regarding aquifer storativity or in absence of pumping tests, change in water table depth does not equate to total water storage fluctuation in an aquifer of uncertain depth and hydraulic characteristics. Second, water flow is governed by hydraulic head rather than water table depth; therefore, a bias in WTD of tens of meters within a continental model that spans thousands of meters of hydraulic head does not necessarily speak to the model's ability to laterally move water through the saturated subsurface.

Finally, perched and confined aquifer systems can completely disconnect anomalies in total subsurface hydrologic stores and measurable WTD fluctuations. However, WTD does indicate vadose zone-saturated zone connectivity, and for unconfined aquifers it is a good indicator for loss or gain in aquifer storage, so we briefly compare observed and simulated WTDs here.

Observed WTD from over 41,000 aquifers across the contiguous United States spans multiple orders of

magnitude and is shown in Fig. 8. The PFCONUSv1 model demonstrates a fairly consistent shallow WTD bias across the domain, with "hot spots" of over 50m depth difference in the southern reaches of the Ogallala aquifer, in the southern Pacific Northwest region, and in the Lower Colorado River basin. However, many of these wells represent locations impacted by extractions (wells are preferentially drilled in regions prioritizing municipal or agricultural groundwater resources), wells tapping confined aquifers, or WT depths that simply cannot be captured

by a shallow aquifer model of 102m depth. In a 1985 transient simulation of PFCONUSv1, Maxwell and Condon (2016, supplemental information) found that while no strong connection exists between water table depth bias and the model's geologic properties, WTD bias was aquifer-dependent, with the greatest positive WTD biases occurring in the High Plains aquifer which has experienced depletions in the last several decades.

Further, WTD is only informative as an indicator of positive or negative $\Delta S_{gw}$ if multiple observations are

provided through time. Therefore, the available USGS wells have been filtered by excluding 1) locations where the observed minimum WTD was greater than 60 m (PFCONUSv1 estimates pressure at cell centers, with the center of the deepest layer at 52 m), 2) locations providing less than 10 observations during the simulation timeframe, 3) locations flagged by the USGS as a confined or mixed aquifer system (aquifer type code aqfr_type_cd in the Groundwater levels for the Nation dataset provided by USGS NWIS, https://waterdata.usgs.gov/nwis/gwlevels/),

and 4) locations flagged for pumping (water level site status code lev_status_cd) during the simulation period.

WTD bias for the remaining 2,486 locations is shown in Fig. 8c. WTD agreement is considerably improved at these locations, but a shallow WTD bias is still present, with 25th, 50th, and 75th quantiles for simulated minus observed difference in total water level being 2.5 m, 5.8 m, and 13.5 m, respectively. However, $\rho$ values suggest that despite PFCONUSv1 shallower water tables, the model is still able to capture temporal fluctuations in depth to

saturation (and by association, groundwater $\Delta S_{gw}$) at almost half of the filtered well sites (Fig. 7d). Quantiles for $\rho$ at the filtered locations are 0.14 (25th), 0.46 (50th), and 0.7 (75th); 46% of gauges show $\rho$ greater than 0.5, 37% of gauges show $\rho$ greater than 0.6, and 25% of gauges show $\rho$ greater than 0.7.





### 3.4.2   Soil moisture, $S_{soil}$

Soil moisture (SM) anomalies, analogous to $S_{soil}$, were compared to the ESA CCI soil moisture product at
0.25° resolution and aggregated to weekly totals. Results are shown in Fig. 9. As in GRACE comparisons, we
compared seasonal amplitude spatial signals across the PFCONUSv1 domain, as well as basin-scale aggregates
through time. The ESA CCI record is a state-of-the-art multi-decadal, global satellite-observation of SM, created
from combining single-sensor active and passive microwave sensors; since its release, the literature has shown good
agreement between the ESA CCI product and spatial and LSM-modeled temporal SM patterns of soil moisture, and
the harmonized product has shown better performance than any of its individual single-sensor inputs (Dorigo et al.,
2017; Gruber et al., 2019). Because we are interested in $\Delta S_{soil}$ over time rather than the total water stored in the soil
at any one moment, comparisons were made to SM anomalies, or relative change in soil moisture with respect to the
mean value.

Broadly speaking, the PFCONUSv1 shows overall lower amplitude in the West and higher amplitude in the
East, relative to the CCI product (Fig. 9a,b). While this could be a result of PFCONUSv1 bias in evapotranspiration
or other fluxes in which seasonal signal is dominant, more likely amplitude differences are simply a result of
temporal coverage or blending algorithms in the ESA CCI product. For instance, for the combined SM product,
blending weights are higher for active microwave sensors in the eastern U.S. and high elevation Rockies, while the
rest of the Southwest and the northern Great Plains region favored passive microwave sensors (Dorigo et al., 2017).
Further, ESA CCI SM is limited by temporal coverage; note that in the majority of the eastern PFCONUSv1
domain, less than 365 observation days are available (most likely a product of high humidity and cloud cover) (Fig.
9b), which makes us less confident in $S_{soil}$ amplitude estimates.

At the aggregated basin scale, however, temporal progression of SM shows good agreement between
PFCONUSv1 and CCI SM for most major basins: Individual $\rho$ values for major basins are 0.25 (Upper Colorado),
0.79 (Lower Colorado), 0.75 (Arkansas-Red-White), 0.75 (Ohio), 0.43 (Missouri), 0.65 (Great Basin), 0.72
(Tennessee) and 0.55 (Upper Mississippi). The very weak correlation in the Upper Colorado basin may be indicative
of large uncertainties in the ESA CCI SM product that have been observed with particular surface conditions: For
regions of dense vegetation, topographic complexity, snow cover or frozen soils, uncertainty in ESA CCI SM is very
high (Dorigo et al., 2017), and we therefore have low confidence in ESACCI comparisons in Rocky Mountain
headwaters regions.

### 3.4.3   Snow water equivalent, $S_{snow}$

Finally, modeled PFCONUSv1 $S_{snow}$ storage component was validated against snow telemetry data in the
mountainous West of the model domain (Fig. 10). An important caveat to note is that point-measured snow water
equivalent (SWE) is likely to consistently overestimate gridded land surface model products, given that coarse-
resolution model cells (in our case, 1km lateral discretization) represent an aggregate of highly heterogenous SWE
and canopy interference across a wide spatial area. Telemetry stations are frequently situated in clearings or in
breaks in canopy density in order to maximize throughfall. For instance, Molotch and Bales (2005) characterized the
distribution of SWE depth in 16-, 4-, and 1-km$^2$ grid elements surrounding SNOTEL stations in Rio Grande
headwaters, using a combination of field observations, remote sensing products, and snowpack mass balance





modeling. They found that in the majority of the sites, the SNOTEL station represented high percentiles of SWE relative to the surround land area, and that SNOTEL site conditions (such as vegetation density, solar radiation index, and terrain indices) were not representative of the vast majority of grid element space. In some regions, SNOTEL SWE was more than 200% greater than the mean grid element value.

As would therefore be expected, the 1-km resolution PFCONUSv1 model underestimates annual peak SWE (water equivalent at maximum accumulation) and April 1 SWE (water equivalent during ablation). PBIAS for annual peak SWE was -50/6%, -33.5%, and -14.7% at 25th, 50th, and 75th percentiles, respectively. April 1 SWE PBIAS was similar, with some individual SNOTEL stations showing more than double the SWE than PFCONUSv1 simulations (Fig. 10c). However, the PFCONUSv1 model clearly reproduces appropriate timing for snow

accumulation and ablation, with the fraction of snow-covered sites tracking almost identically between SNOTEL and PFCONUSv1(Fig. 10d). Percentiles for Spearman's $\rho$ values for cool-season daily SWE (Fig. 10d) are 0.85 (25th percentile), 0.92 (50th), and 0.96 (75th).

### 4    Discussion: Known and unknown sources of model bias

In Sect. 3, outputs from an integrated surface water-groundwater hydrologic model, PFCONUSv1, were compared to available point-scale monitoring networks and remote sensing products in an effort to evaluate the model's ability to reliably reproduce components of the water budget listed in equation (1).

Broadly, results indicate moderate to strong correlation between PFCONUSv1 and point or remote sensing datasets, suggesting very good ability for PFCONUSv1 to simulate continental-scale water balance components.

However, the PFCONUSv1 model should be considered a work in progress; with approximately 31 million cells in the domain, PFCONUSv1 bias can originate from errors associated with model physics, inputs, process representation, or epistemic uncertainty (Table 2). The best publicly available datasets were used to populate and drive this simulation, but such inputs are certainly subject to their own errors and uncertainties and must be continuously revisited to improve their fidelity. In this section, we discuss identifiable errors in model inputs and

implications to future model development.

### 4.1    Meteorological forcing errors and topographic processing

Major biases exist in preprocessing of PFCONUSv1 meteorological forcing and topography, which are peripheral to but act simultaneously with all other sources of bias (Table 2). In this way, isolating the effects of a

single bias source can be challenging. Streamflow itself is sensitive to errors in drainage area, topographic relief, and precipitation or temperature bias, and the errors in surface and subsurface moisture flux can propagate downstream to impact moisture availability and evapotranspiration in areas remote from the original bias source.

### 4.1.1    Terrain processing and drainage area

Topographic slopes were defined from a digital elevation model (DEM) generated by HydroSHEDS and subsequently subjected to a hydrologic processing algorithm, which adjusted drainage networks to remove true and artificial pits, depressions, and barriers and ensure complete river network connectivity (Barnes et al., 2016). However, both loss of resolution in DEMs and the topographic processing can result in loss of topographic relief and





change in drainage area. Therefore, PFCONUSv1 streamflow percent bias should in theory reflect fidelity of
upstream watershed area. We compared PFCONUSv1 drainage area with "true" drainage area determined based on
geospatial stream properties obtained from the GAGES-II (Falcone, 2002).

    Figure 11 shows the relationship between percent difference in observed and simulated streamflow, versus
percent difference in observed and processed drainage area, for all 2,392 USGS stream gauges. There are three
primary conclusions to be drawn from this relationship (Fig. 11a): 1) A clear, linearly proportional correlation exists
between percent difference in drainage and percent difference in streamflow. For streamflow percent difference
from observed ranging from -200 to 200%, we find that 977 out of 2,392 stream gauges fall within ±30% of this
flow-drainage relationship. Essentially, this means that for 41% of gauges in PFCONUSv1, the percent bias in
annual flow can be primarily attributed to errors in topographic processing. 2) A considerable number of gauges
exhibit positive percent difference between observed and simulated annual streamflow, and these gauges typically
are those with very low runoff ratios. Such a finding is not surprising, in that streams with low RR will be
particularly sensitive to external drivers. And 3) a certain amount of noise exists in these drainage-flow
relationships, with many locations exhibited higher or lower error in annual flow than that expected by drainage
errors, regardless of runoff ratio.

    Figure 11b shows locations where the flow-drainage relationship was expected or unexpected. For the
majority of the eastern United States, bias in streamflow is simply a function of drainage area bias from topographic
processing. The mountainous West was considerably noisier, exhibiting in somewhat equal parts lower, higher, or
expected annual flow behavior from drainage bias. We expect that much of the noise in annual flow bias is a
function of precipitation and temperature bias and timing, and subsequently snowpack. However, in the Great Plains
region, the considerable, positive annual flow bias shown in Fig. 3 cannot be attributed to the error in drainage area.
In fact, for 600 gauges in the Great Plains area (~20% of all locations), the percent difference between PFCONUSv1
and true drainage area is near 0, but percent difference in streamflow is between 30 and 200%. We believe that the
greatest driver of this bias is the lack of groundwater extractions in the PFCONUSv1 model. Note that not only is
the PFCONUSv1 model naturalized for the 2002-2006 simulation period, but its initial condition is informed by
1985 naturalized spin-up, which does not include at least 50 years of groundwater depletion. However, some of the
positive annual flow bias behavior in this region could be attributed to some biases in cumulative precipitation,
which is detailed in Sect. 4.1.2.

### 4.1.2 Atmospheric forcing bias

   The NLDAS meteorological forcing, which is described in Sect. 2.2, was bilinearly interpolated across the
PFCONUSv1 domain; biases in precipitation, evaporation, wind speed, humidity, and radiation can therefore come
from either the NLDAS product or its statistical downscaling. We compared daily total precipitation and average
daily air temperature from the interpolated NLDAS product at 9,139 ($P$) and 1,678 (temperature) GHCND
meteorological stations across the PFCONUSv1 domain, calculating relative bias and Spearman's $\rho$ at each location.
Figure 12 summarizes these comparisons. Broadly, we can identify several examples where NLDAS bias are
potential drivers of the bias in timing and volume of hydrologic fluxes:



- PFCONUSv1 Annual precipitation over the Kansas-Nebraska border in the Great Plains region is 10-25% greater than observed (Fig. 12b). This bias could be one source of positive flow bias at USGS stream gauges east of the High Plains aquifer.

- Fidelity in streamflow timing will of course be a function of accurate precipitation timing and intensity. A hydrologic model cannot be expected to perform considerably better than its recharge forcing, or results could be considered spurious. Areas with weakest correlation between observed and NLDAS daily precipitation are in the Rocky Mountain headwaters region (Fig. 12c). In the Upper Colorado watershed as a whole, the $50^{th}$ percentile $\rho$ value for daily precipitation is 0.56, or the lowest of all other major basins. The Upper Colorado is also the basin with poorest overall daily streamflow timing, with $\rho_{50th}$=0.33.

  Similarly, the best performing basin for both streamflow ($\rho_{50th}$=0.63) and precipitation timing ($\rho_{50th}$=0.7) is the Tennessee River Basin.

- Our interpolated NLDAS product underestimates the diurnal temperature fluctuations, primarily by considerably overestimating minimum (nighttime) daily temperature (Fig. 12e), which is likely a considerable driver of underestimated SWE. Further, maximum daily temperature is underestimated over

  the Rockies (Fig. 12h). Given that ET is largely dependent upon available radiative forcing, this could explain some of PFCONUSv1 negative bias at FLUXNET stations over the Rockies.

- Annual temperature errors could also explain several regions where PFCONUSv1 comparisons to the MOD16A2 and to SSEBop MODIS algorithms agree. For example, warm temperature biases and positive ET biases (relative to both MODIS algorithms, Fig. 7g,h) are seen in much of the lower elevations of the

  mountainous West and in the majority of the Pacific Northwest. Spatial patterns of ET biases (Fig. 7g,h) in the Upper Mississippi and Ohio River Basins seem to instead follow the spatial pattern of precipitation bias (Fig. 11b), with regions receiving higher precipitation also experiencing higher ET.

- NLDAS-simulated daily temperature timing is excellent. However, temperature was not deseasonalized before correlation was calculated, and the seasonal signal will certainly account for the majority of

  temperature variance.

More specifically, we can verify specific impacts of NLDAS bias to SWE and ET at individual SNOTEL and FLUXNET sites. Figure 13 shows observed and simulated (or interpolated) meteorological conditions and water balance components for snow and evapotranspiration.

SWE bias at SNOTEL sites is preferentially low at higher elevations (Fig. 13d). While this difference, as discussed above, can to a certain extent be attributed to differences in heterogeneous land and vegetation between the point and grid scale (Molotch and Bales, 2005), we also find that biases in temperature and precipitation likely drive the PFCONUSv1 low bias snowpack. PFCONUSv1 SWE experiences a low bias in cumulative annual precipitation at SNOTEL sites (Fig. 13c), and lower elevations exhibit a warm cool season and annual bias (Fig. 13 a,b), both of which would contribute to low accumulation and high ablation rates.

While NLDAS shows good agreement at FLUXNET locations (Fig. 13g), comparisons between NLDAS and observed vapor pressure deficit and wind speed both exhibit a considerable amount of scatter (Fig. 13e,f).





Specifically, we find that the poorest performing sites for vapor pressure (those in the Upper Colorado river basin) also exhibited the highest magnitude ET biases (Fig. 13h). Lower vapor pressure deficits (0 to 20 Pa) and lower

wind speeds (0 to 6 m s$^{-1}$) have an overall positive bias, which likely contributes to an overall positive ET bias from 0 to 2.5 mm day$^{-1}$ rates. Similarly, we believe bias high-evapotranspiration days (ET > 5 mm day$^{-1}$), which PFCONUSv1 preferentially under-predicts, may be attributed to NLDAS under predicting wind speeds greater than ~10 m s$^{-1}$. Other studies have also highlighted the persistent biases in precipitation and temperature estimates from continental or global meteorological products, which can propagate into hydrologic model predictions (e.g., Ashfaq

et al., 2010; Sperna Weiland et al., 2015).

Errors in atmospheric forcing products often necessitate statistical bias correction before simulations are run (Piani et al., 2010). NDLAS specifically has been validated in its ability to reproduce meteorological conditions for streamflow (Xia, Mitchell, Ek, Cosgrove, et al., 2012), soil moisture (Xia, Ek, et al., 2015), and evapotranspiration (Xia, Hobbins, et al., 2015) prediction by LSMs. While long-term spatial patterns and seasonal

signals were captured for soil moisture and evapotranspiration, results were not particularly encouraging in terms of NLDAS fidelity at daily or weekly timescales. In this study, it is difficult to directly attribute the portion of streamflow, SWE or ET errors that occur from atmospheric forcing bias, but these water balance components would certainly improve with continued progress in meteorological forcing datasets.

**4.2     Anthropogenic process representation and other epistemic errors**

Plenty of sources of uncertainty can contribute to biases not discussed in Sect. 4.1 (Table 2). We have chosen to address meteorological forcing and topographic processing errors above, simply because they are somewhat readily quantifiable, while parameter values and other epistemic uncertainties, such as simplification or scaling of model physics, are poorly constrained or simply unknown. Other biases include population of model

parameter fields.

While we do not discuss model parameter uncertainty, such as conductivity, porosity, van Genuchten parameters, Manning's surface roughness, land and vegetation parameters, or model horizontal and vertical discretization, these are also areas for improvement. For example, Maxwell et al. (2015) show that national geologic and soil parameters datasets are prone to errors via political boundaries, such as state lines; and the PFCONUSv1

model oversimplifies deeper geology, with a 100m vertically homogenous layer.

However, as mentioned above, these fields are poorly constrained at the continental scale, and simulations are necessarily limited by availability of appropriate distributed products. Model parameter population is often addressed through calibration, but population of parameter fields becomes increasingly difficult as resolution increases and calibration becomes more computationally demanding. Future model iterations may need to take

advantage of methods that allow transfer of parameters (e.g. conductivity) from coarse-resolution, efficient models to high resolution ones (Foster & Maxwell, 2019). Model discretization is another concern. Coarsening of vertical and lateral resolution is a necessity at the continental scale but aggregating to ~1km resolution certainly comes with inherent uncertainties and loss of information. DEMs in particular will lose topographic drivers with scale (Wu et al., 2008), resulting in loss of relief; but on the other hand, coarse resolution cells could result in inappropriately





steep pressure gradients as a function of Richards' equation parameterization and pressure-dependent permeability (Maxwell and Condon, 2016), and suitable vertical length scales for Richards' equations generally do not exceed several meters (Or et al., 2015). This calls to question the scalability of model physics; as Beven and Cloke (2012) rightly point out, whether or not governing partial differential equations will scale linearly is a concern. However, the current governing equations for PFCONUSv1 are simply the best currently known representation of hydrologic

processes at this scale.

Finally, process representation is certainly a concern. Transient anthropogenic modules, such as urban hydrology models, pumping and injections, or surface water diversions, are currently possible but add to computational demand and require detailed historical data on water use with temporal and spatial coverage simply not yet available. As a naturalized model, the PFCONUSv1 simulations presented here will necessarily overpredict

water tables and baseflow in regions where extractions are apparent. For instance, Maxwell and Condon (2016, supplemental information), show streamflow examples at Lees Ferry USGS gauge, where timing and volume of streamflow are entirely governed by dam hydraulics. Condon and Maxwell (2019) show that incorporating a century of groundwater depletion across the PFCONUSv1 domain considerably decreases streamflow, with sensitivity to pumping concentrating downstream; more specifically, they found that long-term depletions over the High Plains

aquifer resulted in a swap of discharging to recharging groundwater. However, naturalized continental models with high fidelity in non-anthropogenic settings could be used to estimate impact from human influence, simply by examining the difference between observed and simulated conditions.

## 5      Conclusions and implications

In this study, we present the detailed validation of a transient, coupled hydrologic-land surface simulation at the continental scale and at hyper-resolution using a diverse set of monitoring networks and state-of-the-art remote sensing products. We found that PFCONUSv1 produced very good temporal patterns. The following are 50th percentile (in space, over the entire domain) Spearman's rank correlation $\rho$ for individual water balance components: $\rho_{50th} = 0.65$ for $R$, with evaluation against daily USGS stream gauge observations; $\rho_{50th} = 0.72$ for $ET$,

with evaluation against daily FLUXNET eddy covariance observations (for monthly HUC8-aggregated remote sensing products, $\rho_{50th} = 0.85$ for $ET$ relative to MOD16A2 algorithm and $\rho_{50th} = 0.91$ for $ET$ relative to SSEBop algorithm); $\rho_{50th} = 0.80$ for major basin-aggregate $S$, with evaluation against monthly GRACE remote gravity field sensing; $\rho_{50th} = 0.46$ for filtered USGS well observations, which are related but not equivalent to $S_{gw}$; $\rho_{50th} = 0.69$ major basin-aggregate $S_{soil}$, with evaluation against ESA CCI weekly SM from active/passive microwave sensors

(Upper Colorado is not excluded, but note the uncertainties in ESA CCI over snow-covered, densely forested, and topographically complex land area); and $\rho_{50th} = 0.92$ for $S_{snow}$, with evaluation against daily SNOTEL point observations. Further, we discussed three primary sources of model bias: terrain processing, errors in atmospheric forcing, and lack of anthropogenic influence.

The results presented here provide benefits to the high-resolution, continental (and above) -scale hydrologic

community. First, our generally good agreement with observations and remote sensing products suggests great promise for extreme-scale and high-resolution modeling to become a reality. We argue that PFCONUSv1 and





similar models are feasible and will certainly see improvements in the near future with increased availability of open-access and distributed datasets, remote sensing advancements, improved monitoring networks, and advancements in highly parallelized computing.

Second, these results provide a benchmark for PFCONUS development. Some areas for model improvement that were immediately identified in this study include the following: 1) The source of high positive bias in the Central Plains should be further addressed. While we propose that this bias is largely attributed to the lack of groundwater pumping in the model (we estimate that 25% of stream gauges are impacted by High Plains aquifer depletions), other potential sources of error could include inappropriate soil or geology hydraulic conductivity or

van Genuchten parameters, the lack of spatially distributed Manning's coefficient (Maxwell et al., 2015), or loss of topographic relief associated with 1-km lateral resolution. 2) We show that topographic processing has resulted in considerable error in drainage area for approximately 40% of stream gauges. Accessible improvements could be made to streamflow bias with improved topographic processing algorithms. And, 3) interpolated atmospheric forcing from NLDAS reanalysis has two primary biases that, if corrected with statistical bias correction, would

immediately benefit streamflow, ET, and snow water equivalent. First, precipitation timing is lacking in many areas of the domain, particularly over the Rocky Mountain region. Second, mean nighttime air temperature exhibits a high temperature bias, resulting in an underestimated diurnal temperature fluctuation for the majority of the domain. Average daytime maximum temperature is also underestimated over the Rocky Mountains.

Finally, we argue that model fidelity can only be reliably understood at a process-based level if all water

balance components available in the model outputs are evaluated. While single parameter validation may be effective for operational forecasts, we do risk the equifinality dilemma of arriving at the right answer for the wrong reasons (Kirchner, 2006). The value in the type of validation exercise presented here is clearly a mechanistic understanding of model performance and a higher level of confidence in overall water balance modeling skill. Further work should be done to continue to incorporate additional observational and remote sensing networks as

they become available. Impressive model validation toolkits that exist in the land surface community, such as the Land surface Validation Toolkit (LVT) (Kumar et al., 2012) and the International Land Model Benchmarking (ILAMB) System (Collier et al., 2018), are inspiring collaborative efforts to streamline and standardize model evaluation. We hope to take advantage of these platforms in the future, to compare model performance to other similar continental- and global-scale simulations, to standardize our model evaluation, and to add to our existing

validation datasets. Further work is also needed to assess the scale gaps prevalent in observation and remote sensing data. Specifically, point-scale observations sensitive to small-scale heterogeneity, such as in-situ soil moisture observations, are unlikely to be applicable to the 1-km scale; conversely, we cannot guarantee that PFCONUSv1 outputs scale linearly to coarser-resolution products and models. Improved understanding of how model bias scales with loss of spatial or temporal resolution is a vital area of research.






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

        Proof of concept of regional scale hydrologic simulations at hydrologic resolution utilizing massively parallel
        computer resources. *Water Resources Research*, *46*(4). https://doi.org/10.1029/2009WR008730

Kumar, S. V, Peters-Lidard, C. D., Santanello, J., Harrison, K., Liu, Y., & Shaw, M. (2012). Geoscientific Model
        Development Land surface Verification Toolkit (LVT)-a generalized framework for land surface model
        evaluation. *Geosci. Model Dev*, *5*, 869–886. https://doi.org/10.5194/gmd-5-869-2012

Landerer, F. W., & Swenson, S. C. (2012). Accuracy of scaled GRACE terrestrial water storage estimates. *Water
        Resources Research*, *48*(4). https://doi.org/10.1029/2011WR011453

Lehner, B., Verdin, K., & Jarvis, A. (2008). New Global Hydrography Derived From Spaceborne Elevation Data.
        *Eos, Transactions American Geophysical Union*, *89*(10), 93. https://doi.org/10.1029/2008EO100001

M. Foster, L., & M. Maxwell, R. (2019). Sensitivity analysis of hydraulic conductivity and Manning's *n* parameters
        lead to new method to scale effective hydraulic conductivity across model resolutions. *Hydrological
        Processes*, *33*(3), 332–349. https://doi.org/10.1002/hyp.13327

Maurer, E. P., Wood, A. W., Adam, J. C., Lettenmaier, D. P., Nijssen, B., Maurer, E. P., Wood, A. W., Adam, J. C.,
        Lettenmaier, D. P., & Nijssen, B. (2002). A Long-Term Hydrologically Based Dataset of Land Surface Fluxes
        and States for the Conterminous United States*. *Http://Dx.Doi.Org/10.1175/1520-
        0442(2002)015<3237:ALTHBD>2.0.CO;2*. https://doi.org/10.1175/1520-
        0442(2002)015<3237:ALTHBD>2.0.CO;2

Maxwell, R. M., Condon, L. E., & Kollet, S. J. (2015). A high-resolution simulation of groundwater and surface
        water over most of the continental US with the integrated hydrologic model ParFlow v3. *Geoscientific Model
        Development*, *8*(3), 923–937. https://doi.org/10.5194/gmd-8-923-2015

Maxwell, Reed M. (2013). A terrain-following grid transform and preconditioner for parallel, large-scale, integrated
        hydrologic modeling. *Advances in Water Resources*, *53*, 109–117.
https://doi.org/10.1016/J.ADVWATRES.2012.10.001

Maxwell, Reed M., & Miller, N. L. (2005). Development of a Coupled Land Surface and Groundwater Model.
        *Journal of Hydrometeorology*, *6*(3), 233–247. https://doi.org/10.1175/JHM422.1

Maxwell, Reed M, & Condon, L. E. (2016). Connections between groundwater flow and transpiration partitioning.
        *Science (New York, N.Y.)*, *353*(6297), 377–380. https://doi.org/10.1126/science.aaf7891

Maxwell, Reed M, Condon, L. E., Kollet, S. J., Maher, K., Haggerty, R., & Forrester, M. M. (2016). The imprint of
        climate and geology on the residence times of groundwater. *Geophysical Research Letters*, *43*(2), 701–708.





https://doi.org/10.1002/2015GL066916

McGuire, V. L., Lund, K. D., & Densmore, B. K. (1980). *Saturated Thickness and Water in Storage in the High Plains Aquifer, 2009, and Water-Level Changes and Changes in Water in Storage in the High Plains Aquifer, 1980 to 1995, 1995 to 2000, 2000 to 2005, and 2005 to 2009.*

Molotch, N. P., & Bales, R. C. (2005). Scaling snow observations from the point to the grid element: Implications for observation network design. *Water Resources Research*, *41*(11), 1–16. https://doi.org/10.1029/2005WR004229

Moriasi, D. N., Arnold, J. G., Liew, M. W. Van, Bingner, R. L., Harmel, R. D., & Veith, T. L. (1983). MODEL EVALUATION GUIDELINES FOR SYSTEMATIC QUANTIFICATION OF ACCURACY IN WATERSHED SIMULATIONS. In *Transactions of the ASABE* (Vol. 50, Issue 3).

Mu, Q., Heinsch, F. A., Zhao, M., & Running, S. W. (2007). Development of a global evapotranspiration algorithm based on MODIS and global meteorology data. *Remote Sensing of Environment*, *111*(4), 519–536. https://doi.org/10.1016/j.rse.2007.04.015

Mu, Q., Zhao, M., & Running, S. W. (2011). Improvements to a MODIS global terrestrial evapotranspiration algorithm. *Remote Sensing of Environment*, *115*(8), 1781–1800. https://doi.org/10.1016/j.rse.2011.02.019

Niraula, R., Meixner, T., Ajami, H., Rodell, M., Gochis, D., & Castro, C. L. (2017). Comparing potential recharge estimates from three Land Surface Models across the western US. *Journal of Hydrology*, *545*, 410–423. https://doi.org/10.1016/j.jhydrol.2016.12.028

Niu, G. Y., Yang, Z. L., Mitchell, K. E., Chen, F., Ek, M. B., Barlage, M., Kumar, A., Manning, K., Niyogi, D., Rosero, E., Tewari, M., & Xia, Y. (2011). The community Noah land surface model with multiparameterization options (Noah-MP): 1. Model description and evaluation with local-scale measurements. *Journal of Geophysical Research Atmospheres*, *116*(12). https://doi.org/10.1029/2010JD015139

*Office of Water Prediction*. (n.d.). Retrieved March 23, 2020, from https://water.noaa.gov/about/nwm

Oki, T., & Kanae, S. (2006). Global hydrological cycles and world water resources. *Science (New York, N.Y.)*, *313*(5790), 1068–1072. https://doi.org/10.1126/science.1128845

Or, D., Lehmann, P., & Assouline, S. (2015). Natural length scales define the range of applicability of the Richards equation for capillary flows. *Water Resources Research*, *51*(9), 7130–7144. https://doi.org/10.1002/2015WR017034

Osei-Kuffuor, D., Maxwell, R. M., & Woodward, C. S. (2014). Improved numerical solvers for implicit coupling of subsurface and overland flow. *Advances in Water Resources*, *74*, 185–195. https://doi.org/10.1016/j.advwatres.2014.09.006

Pan, M., Cai, X., Chaney, N. W., Entekhabi, D., & Wood, E. F. (2016). An initial assessment of SMAP soil moisture retrievals using high-resolution model simulations and in situ observations. *Geophysical Research Letters*, *43*(18), 9662–9668. https://doi.org/10.1002/2016GL069964

Piani, C., Weedon, G. P., Best, M., Gomes, S. M., Viterbo, P., Hagemann, S., & Haerter, J. O. (2010). Statistical bias correction of global simulated daily precipitation and temperature for the application of hydrological



models. *Journal of Hydrology*, *395*(3–4), 199–215. https://doi.org/10.1016/j.jhydrol.2010.10.024

Rakovec, O., Mizukami, N., Kumar, R., Newman, A. J., Thober, S., Wood, A. W., Clark, M. P., & Samaniego, L.
(2019). Diagnostic Evaluation of Large-Domain Hydrologic Models Calibrated Across the Contiguous United
States. *Journal of Geophysical Research: Atmospheres*, *124*(24), 13991–14007.
https://doi.org/10.1029/2019JD030767

Reynolds, O. (n.d.). On the Dynamical Theory of Incompressible Viscous Fluids and the Determination of the
Criterion. In *Philosophical Transactions of the Royal Society of London. A* (Vol. 186, pp. 123–164). Royal
Society. https://doi.org/10.2307/90643

Rodell, M., McWilliams, E. B., Famiglietti, J. S., Beaudoing, H. K., & Nigro, J. (2011). Estimating
evapotranspiration using an observation based terrestrial water budget. *Hydrological Processes*, *25*(26), 4082–
4092. https://doi.org/10.1002/hyp.8369

Salas, F. R., Somos-Valenzuela, M. A., Dugger, A., Maidment, D. R., Gochis, D. J., David, C. H., Yu, W., Ding, D.,
Clark, E. P., & Noman, N. (2018). Towards Real-Time Continental Scale Streamflow Simulation in
Continuous and Discrete Space. *JAWRA Journal of the American Water Resources Association*, *54*(1), 7–27.
https://doi.org/10.1111/1752-1688.12586

Save, H., Bettadpur, S., & Tapley, B. D. (2016). High-resolution CSR GRACE RL05 mascons. *Journal of
Geophysical Research: Solid Earth*, *121*(10), 7547–7569. https://doi.org/10.1002/2016JB013007

Schmied, H. M., Cáceres, D., Eisner, S., Flörke, M., Herbert, C., Niemann, C., Peiris, T. A., Popat, E., Portmann, F.
T., Reinecke, R., Schumacher, M., Shadkam, S., Telteu, C.-E., Trautmann, T., & Döll, P. (2020). The global
water resources and use model WaterGAP v2.2d: Model description and evaluation. *Geoscientific Model
Development*. https://doi.org/10.5194/gmd-2020-225

Senay, G B, Gowda, P. H., Bohms, S., Howell, T. A., Friedrichs, M., Marek, T. H., & Verdin, J. P. (2014).
Evaluating the SSEBop approach for evapotranspiration mapping Evaluating the SSEBop approach for
evapotranspiration mapping with landsat data using lysimetric observations in the semi-arid Texas High Plains
Evaluating the SSEBop approach for evapotranspiration mapping. *Hydrol. Earth Syst. Sci. Discuss*, *11*, 723–
756. https://doi.org/10.5194/hessd-11-723-2014

Senay, Gabriel B., Bohms, S., Singh, R. K., Gowda, P. H., Velpuri, N. M., Alemu, H., & Verdin, J. P. (2013).
Operational Evapotranspiration Mapping Using Remote Sensing and Weather Datasets: A New
Parameterization for the SSEB Approach. *JAWRA Journal of the American Water Resources Association*,
*49*(3), 577–591. https://doi.org/10.1111/jawr.12057

Sperna Weiland, F. C., Vrugt, J. A., van Beek, R. L. P. H., Weerts, A. H., & Bierkens, M. F. P. (2015). Significant
uncertainty in global scale hydrological modeling from precipitation data errors. *Journal of Hydrology*, *529*,
1095–1115. https://doi.org/10.1016/j.jhydrol.2015.08.061

Strassberg, G., Scanlon, B. R., & Chambers, D. (2009). Evaluation of groundwater storage monitoring with the
GRACE satellite: Case study of the High Plains aquifer, central United States. *Water Resources Research*,
*45*(5). https://doi.org/10.1029/2008WR006892

Sutanudjaja, E. H., van Beek, R., Wanders, N., Wada, Y., Bosmans, J. H. C., Drost, N., van der Ent, R. J., de Graaf,





I. E. M., Hoch, J. M., de Jong, K., Karssenberg, D., López López, P., Peßenteiner, S., Schmitz, O., Straatsma, M. W., Vannametee, E., Wisser, D., & Bierkens, M. F. P. (2018). PCR-GLOBWB 2: a 5 arcmin global hydrological and water resources model. *Geoscientific Model Development*, *11*(6), 2429–2453. https://doi.org/10.5194/gmd-11-2429-2018

Swinbank, W. C., & Swinbank, W. C. (1951). THE MEASUREMENT OF VERTICAL TRANSFER OF HEAT AND WATER VAPOR BY EDDIES IN THE LOWER ATMOSPHERE. *Journal of Meteorology*, *8*(3), 135–145. https://doi.org/10.1175/1520-0469(1951)008<0135:TMOVTO>2.0.CO;2

Tang, Q., Gao, H., Yeh, P., Oki, T., Su, F., & Lettenmaier, D. P. (2010). Dynamics of Terrestrial Water Storage Change from Satellite and Surface Observations and Modeling. *Journal of Hydrometeorology*, *11*(1), 156–
170. https://doi.org/10.1175/2009JHM1152.1

Troch, P., Durcik, M., Seneviratne, S., Hirschi, M., Teuling, A., Hurkmans, R., & Hasan, S. (2007). New data sets to estimate terrestrial water storage change. *Eos, Transactions American Geophysical Union*, *88*(45), 469–470. https://doi.org/10.1029/2007EO450001

van Genuchten, M. T. (1980). A Closed-form Equation for Predicting the Hydraulic Conductivity of Unsaturated
Soils1. *Soil Science Society of America Journal*, *44*(5), 892. https://doi.org/10.2136/sssaj1980.03615995004400050002x

Velpuri, N. M., Senay, G. B., Singh, R. K., Bohms, S., & Verdin, J. P. (2013). A comprehensive evaluation of two MODIS evapotranspiration products over the conterminous United States: Using point and gridded FLUXNET and water balance ET. *Remote Sensing of Environment*, *139*, 35–49.
https://doi.org/10.1016/j.rse.2013.07.013

Westerhoff, R. S. (2015). Using uncertainty of Penman and Penman-Monteith methods in combined satellite and ground-based evapotranspiration estimates. *Remote Sensing of Environment*, *169*, 102–112. https://doi.org/10.1016/j.rse.2015.07.021

Wiese, D. N., Landerer, F. W., & Watkins, M. M. (2016). Quantifying and reducing leakage errors in the JPL
RL05M GRACE mascon solution. *Water Resources Research*, *52*(9), 7490–7502. https://doi.org/10.1002/2016WR019344

Wilson, K. B., Hanson, P. J., Mulholland, P. J., Baldocchi, D. D., & Wullschleger, S. D. (2001). A comparison of methods for determining forest evapotranspiration and its components: sap-flow, soil water budget, eddy covariance and catchment water balance. *Agricultural and Forest Meteorology*, *106*(2), 153–168.
https://doi.org/10.1016/S0168-1923(00)00199-4

Wood, E. F., Roundy, J. K., Troy, T. J., van Beek, L. P. H., Bierkens, M. F. P., Blyth, E., de Roo, A., Döll, P., Ek, M., Famiglietti, J., Gochis, D., van de Giesen, N., Houser, P., Jaffé, P. R., Kollet, S., Lehner, B., Lettenmaier, D. P., Peters-Lidard, C., Sivapalan, M., … Whitehead, P. (2011). Hyperresolution global land surface modeling: Meeting a grand challenge for monitoring Earth's terrestrial water. *Water Resources Research*,
*47*(5). https://doi.org/10.1029/2010WR010090

Wu, S., Li, J., & Huang, G. H. (2008). A study on DEM-derived primary topographic attributes for hydrologic applications: Sensitivity to elevation data resolution. *Applied Geography*, *28*(3), 210–223.



https://doi.org/10.1016/j.apgeog.2008.02.006

Xia, Y., Ek, M. B., Wu, Y., Ford, T., & Quiring, S. M. (2015). Comparison of NLDAS-2 Simulated and NASMD Observed Daily Soil Moisture. Part I: Comparison and Analysis. *Journal of Hydrometeorology*, *16*(5), 1962–1980. https://doi.org/10.1175/JHM-D-14-0096.1

Xia, Y., Hobbins, M. T., Mu, Q., & Ek, M. B. (2015). Evaluation of NLDAS-2 evapotranspiration against tower flux site observations. *Hydrological Processes*, *29*(7), 1757–1771. https://doi.org/10.1002/hyp.10299

Xia, Y., Mitchell, K., Ek, M., Cosgrove, B., Sheffield, J., Luo, L., Alonge, C., Wei, H., Meng, J., Livneh, B., Duan, Q., & Lohmann, D. (2012). Continental-scale water and energy flux analysis and validation for North American Land Data Assimilation System project phase 2 (NLDAS-2): 2. Validation of model-simulated streamflow. *Journal of Geophysical Research: Atmospheres*, *117*(D3), n/a-n/a. https://doi.org/10.1029/2011JD016051

Xia, Y., Mitchell, K., Ek, M., Sheffield, J., Cosgrove, B., Wood, E., Luo, L., Alonge, C., Wei, H., Meng, J., Livneh, B., Lettenmaier, D., Koren, V., Duan, Q., Mo, K., Fan, Y., & Mocko, D. (2012). Continental-scale water and energy flux analysis and validation for the North American Land Data Assimilation System project phase 2 (NLDAS-2): 1. Intercomparison and application of model products. *Journal of Geophysical Research: Atmospheres*, *117*(D3), n/a-n/a. https://doi.org/10.1029/2011JD016048

Xu, C. Y., & Singh, V. P. (2005). Evaluation of three complementary relationship evapotranspiration models by water balance approach to estimate actual regional evapotranspiration in different climatic regions. *Journal of Hydrology*, *308*(1–4), 105–121. https://doi.org/10.1016/j.jhydrol.2004.10.024

Zaitchik, B. F., Rodell, M., & Olivera, F. (2010). Evaluation of the Global Land Data Assimilation System using global river discharge data and a source-to-sink routing scheme. *Water Resources Research*, *46*(6). https://doi.org/10.1029/2009WR007811

Zhang, L., Dobslaw, H., Stacke, T., Güntner, A., Dill, R., & Thomas, M. (2017). Validation of terrestrial water storage variations as simulated by different global numerical models with GRACE satellite observations. *Hydrol. Earth Syst. Sci*, *21*, 821–837. https://doi.org/10.5194/hess-21-821-2017





Table 1: Products used to evaluate PFCONUSv1 simulated water balance component performance

| Water balance component | Data product | Spatial scale | Product type |
|---|---|---|---|
| *Comparisons to Modeled Water Budget Components* | | | |
| Surface runoff, $R$ | USGS stream gauges | Aggregate of upstream area, 2392 locations | Point observation |
| Evapotranspiration, $ET$ | MODIS | 1 km resolution, global scale | Remote sensing |
| | SSeBOP | 1 km resolution, global scale | Remote sensing |
| | FLUXNET | Local, 30 locations | Point observation |
| Storage, $\Delta S$ | SNOTEL | Local, 556 locations | Point observation |
| | GRACE (5 products) | 0.25 to 1 degree resolution, 3 degree basis function, global scale | Remote sensing |
| | ESACCI Active/Passive | 0.25 degree resolution, global scale | Remote sensing |
| | USGS wells | Local 41,269 locations static, 2486 locations temporal | Point observation |
| *Comparisons to atmospheric forcing* | | | |
| Precipitation, Temperature | GHCND | Local, 9139 locations | Point observation |
| Precipitation, Temperature | SNOTEL | Local, 556 locations | Point observation |
| Temperature, Vapor Pressure Deficit, Wind Speed | FLUXNET | Local, 30 locations | Point observation |




Table 2: Examples of potential sources of bias acting on PFCONUSv1 results

| Bias category | Bias source examples | Components directly affected |
|---|---|---|
| *Topographic processing* | Watershed drainage area | Surface flow volume |
| | Topographic relief | Surface flow volume and timing |
| | Stream network mapping | Surface flow volume and timing |
| *Atmospheric forcing* | Precipitation volume | Surface flow volume and SWE |
| | Precipitation timing and intensity | Storm hydrographs |
| | Temperature trends and diurnal, seasonal cycles | Evapotranspiration, snowmelt amount and timing |
| | Humidity | Evapotranspiration |
| | Wind speed | Evapotranspirtion |
| *Anthropogenic* | Dams and reservoirs | Surface flow volume and timing |
| | Groundwater extractions | Groundwater storage |
| | Land disturbance | Evapotranspiration, snow accumulation |
| *Model parameters* | Hydraulic conductivity | Infiltration, recharge |
| | Porosity | Subsurface storage |
| | Manning's $n$ | Surface flow timing and hydrograph |
| | Land and vegetation (albedo, LAI) | Evapotranspiration, snow accumulation and melt |
| | Aquifer model depth | Groundwater storage |
| | Initial conditions of pressure and saturation | Groundwater depth |
| *Epistemic uncertainty* | Scalability of model physics | All/unknown |
| | Vertical and lateral parameter aggregation | |
| | Process interaction; groundwater-surface water and land-atmosphere exchange at various spatial and temporal scales | |



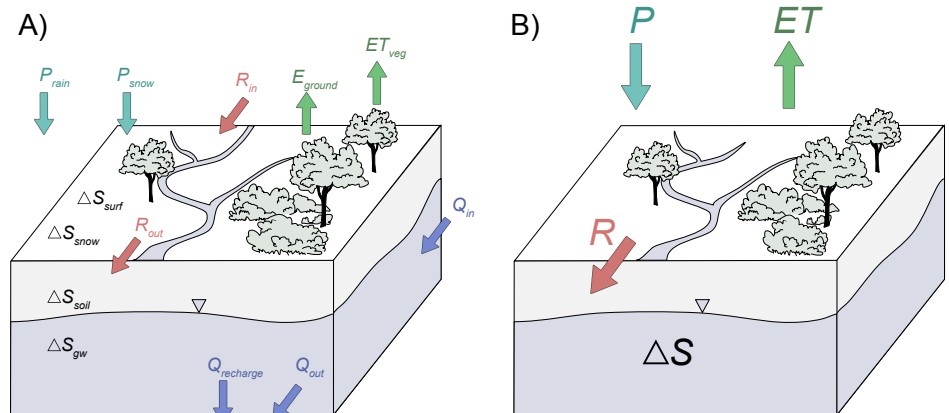

Figure 1: A conceptual model of the a) complete and b) simplified water budget for a hydrologic control volume, corresponding to equations (1) and (2).




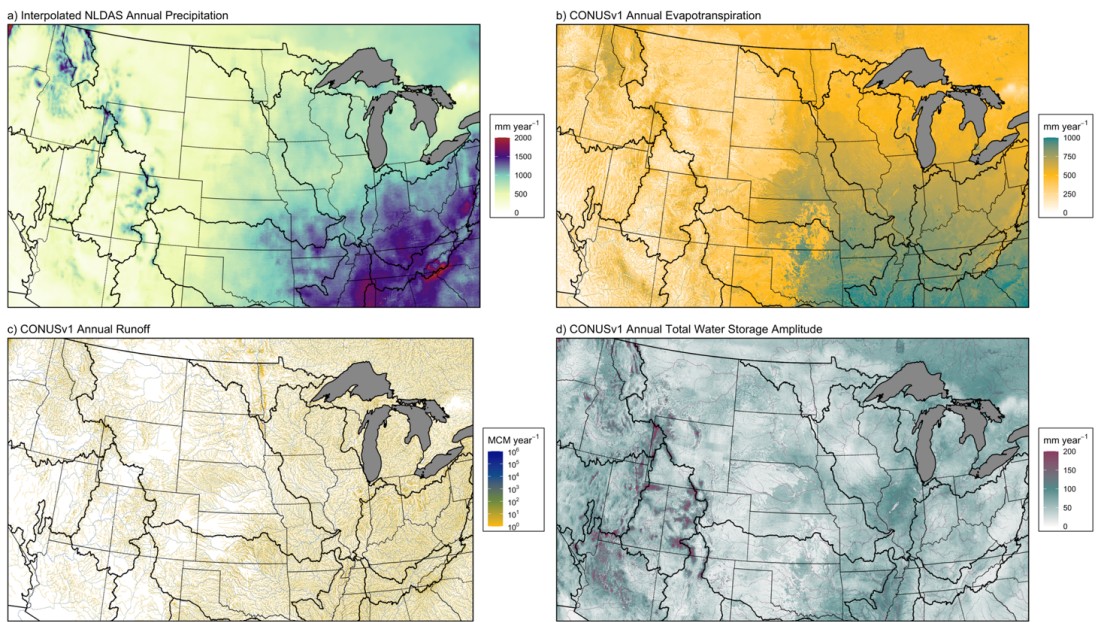

Figure 2: Mean annual water balance components from PFCONUSv1 at 1-km resolution: a) interpolated precipitation *P* from atmospheric forcing inputs, b) simulated mean annual evapotranspiration *ET*, c) simulated mean annual runoff *R*, and d) simulated mean annual total water storage *ΔS* amplitude (combined seasonality of snow water equivalent, groundwater, soil water, and surface water). Total water storage amplitude is the peak-to-peak seasonal storage anomaly, rather than annual storage trend; seasonality (rather than interannual variability) explained the majority of the variance in total *ΔS*. Dotted lines are states, while thicker solid lines are major U.S. river basin outlines, which are labeled in (a).



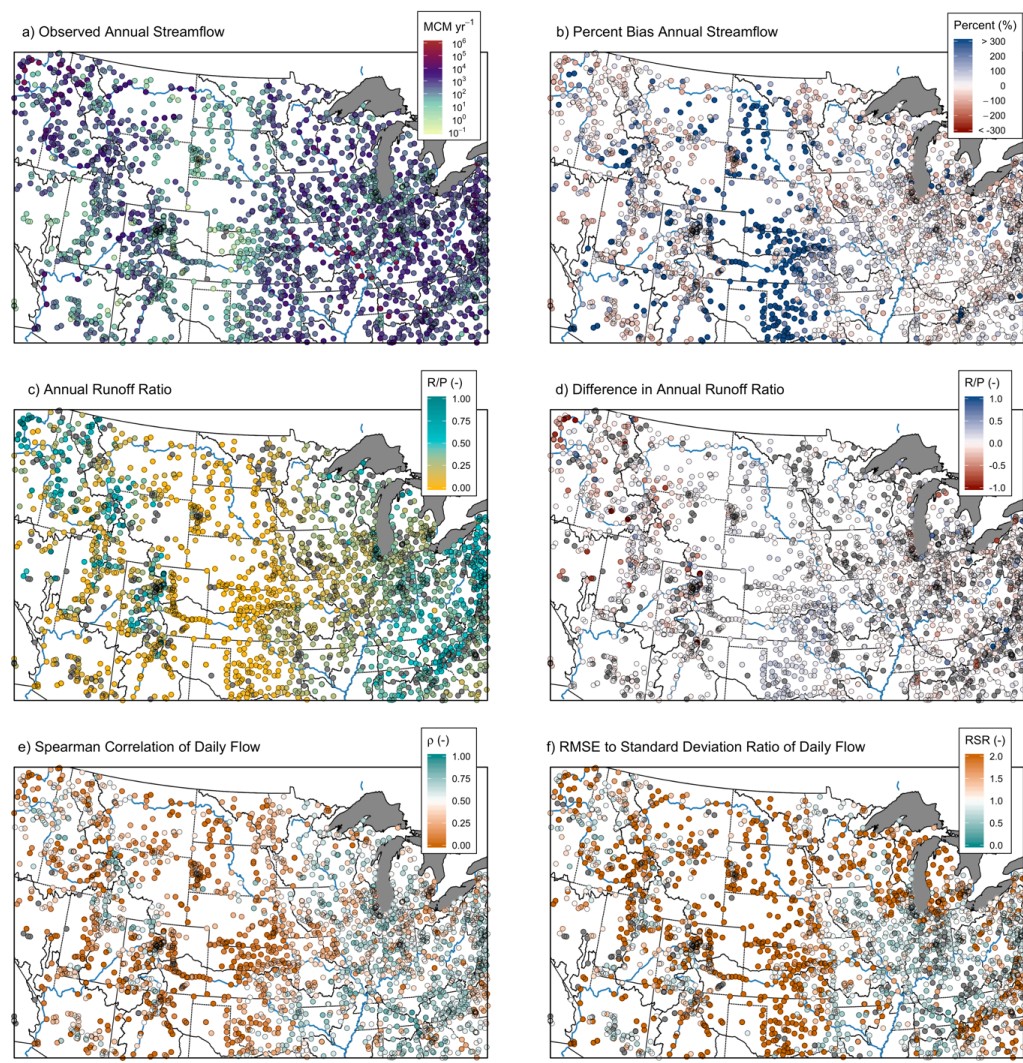

Figure 3: (a) Observed annual streamflow $R$ from USGS gauge network, (b) PBIAS for simulated PFCONUSv1 streamflow, (c) runoff ratio calculated from USGS stream gauges and GHCND precipitation gauges, (d) simulated minus observed runoff ratio, (e) $\rho$ of simulated daily flows, and (f) RSR of simulated daily flows.




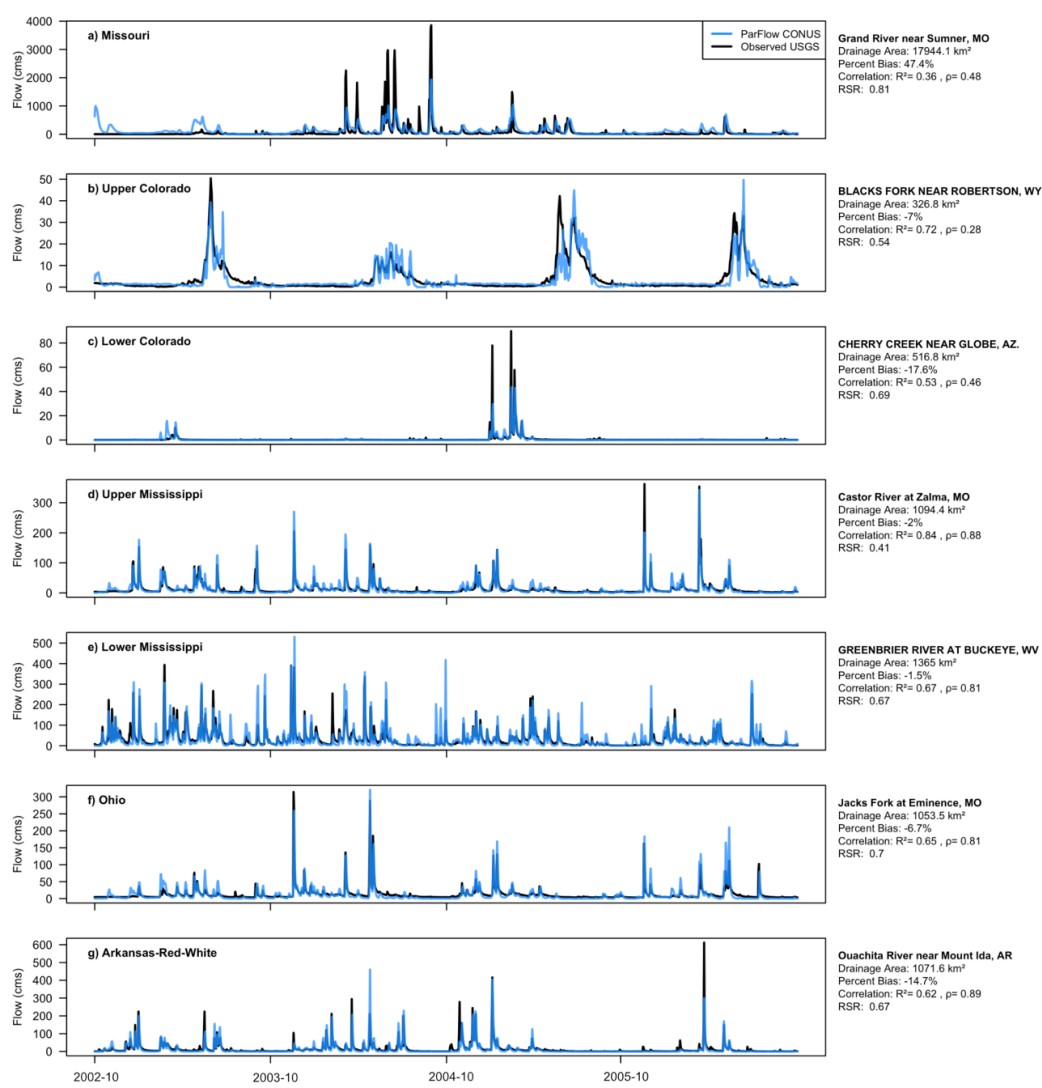

Figure 4: Timeseries of PFCONUSv1 modeled and USGS observed streamflow timeseries at representative gauge locations for each major basin.





 Figure 5: a) Cumulative annual ET observed at 30 FLUXNET sites across the contiguous United States, b) percent bias of PFCONUSv1 daily simulated ET at FLUXNET locations, c) Spearman $\rho$ of PFCONUSv1 daily simulated ET at FLUXNET locations, and d) RSR of PFCONUSv1 simulated daily ET, and e-g) examples of observed and simulated daily ET at three FLUXNET sites with complete observation periods during the simulation timeframe.




Figure 6: PFCONUSv1 ET estimates compared to results from MODIS remote sensing and thermal imaging algorithms. a-c) Annual cumulative ET across HUC8 watersheds, d-f) differences in annual ET, g-i) PBIAS of monthly ET, j-l) Spearman's $\rho$ of monthly ET, and m-o) RSR of monthly ET, for PFCONUSv1 and MODIS products (MOD16A2 and SSEBop algorithms).


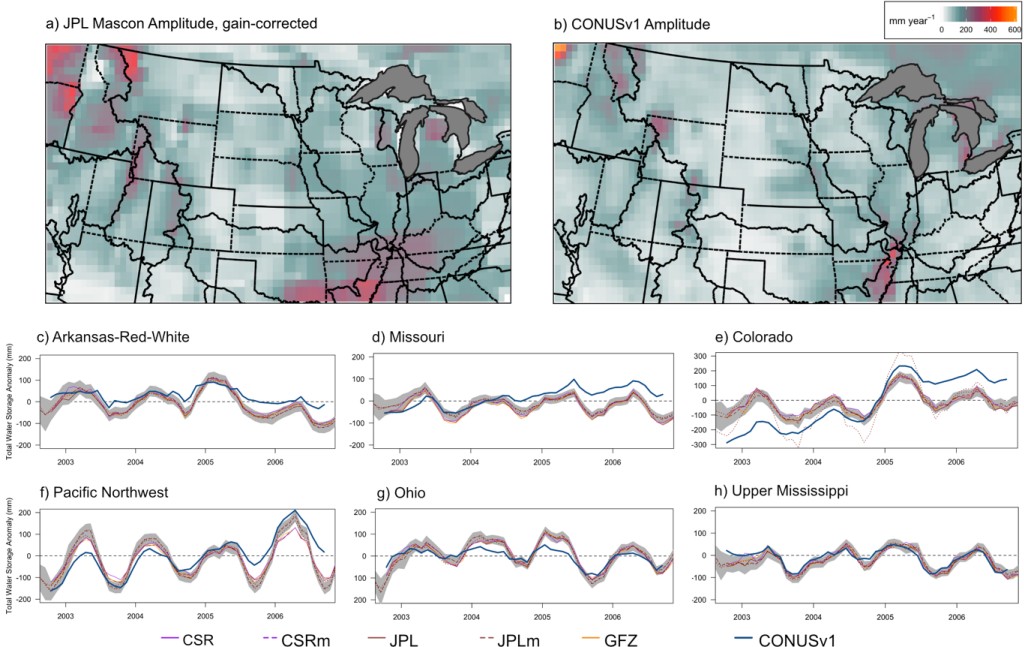

Figure 7: Summary of GRACE and PFCONUSv1 comparisons. Seasonal storage amplitude for a) the JPL mascon solution and b) PFCONUSv1 total water storage, with darker red areas indicating a high degree of seasonality and white areas indicating no sub-annual storage fluctuation. c-h) Timeseries of total water storage anomalies for five GRACE products and for PFCONUSv1 across complete major basins in the PFCONUSv1 domain. Shaded regions indicate uncertainty in the JPLm product based on leakage and measurement error.


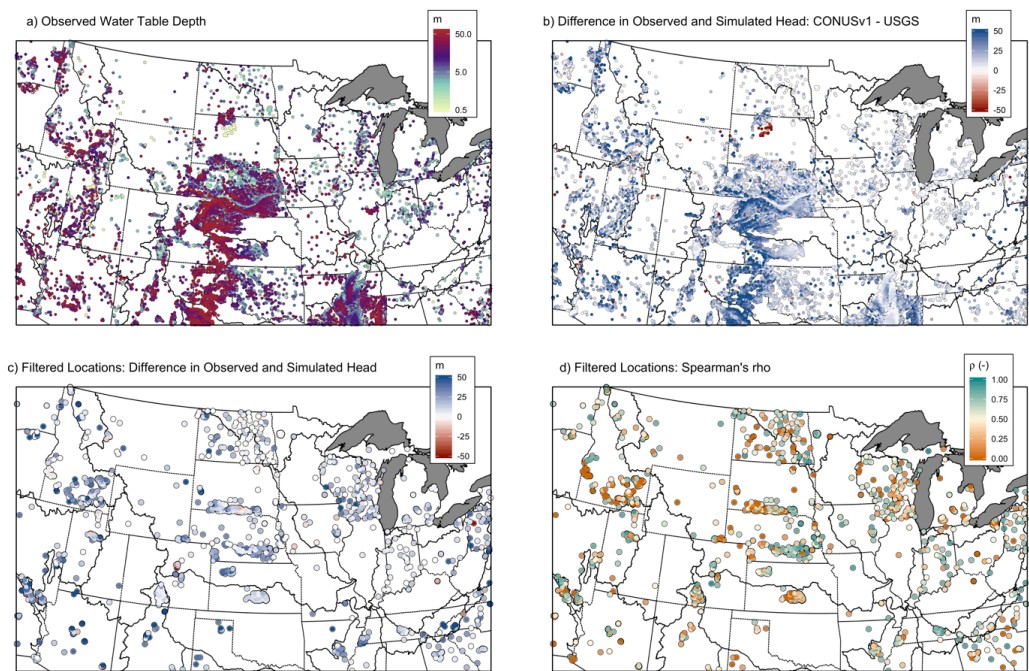

Figure 8: a) Observed water table depth (N=41,269), b) difference in observed and PFCONUSv1 simulated WTD, c) difference in observed and PFCONUSv1 simulated WTD at filtered locations (N=2486), and d) Spearman $\rho$ values at filtered locations using at least 10 instantaneous (daily) observations.




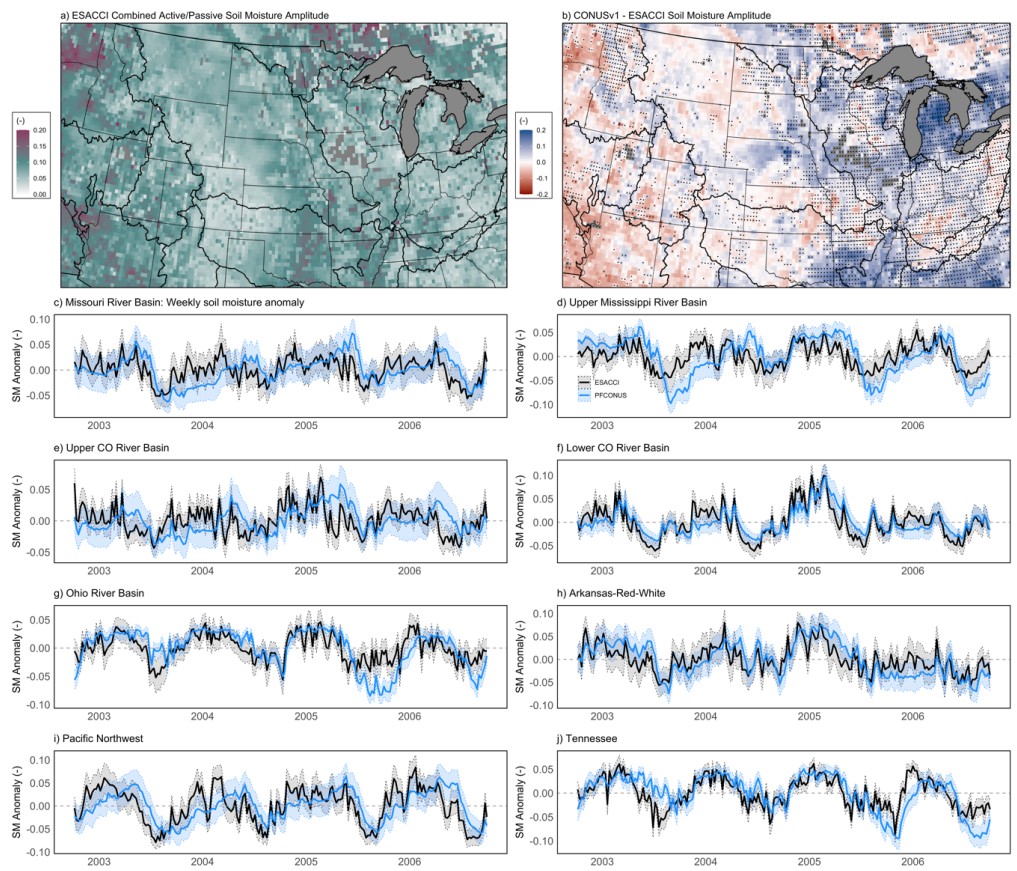

Figure 9: Summary of ESACCI and PFCONUSv1 soil moisture comparisons. Seasonal SM amplitude for a) the
ESACCI mascon solution and b) PFCONUSv1 – ESACCI amplitude difference. Stippling in (b) indicates < 365
days of SM observations available for the ESACCI product, while grey areas (excluded) indicate <1 month of
available data. c-h) Timeseries of weekly SM anomalies across complete major basins in the CONUSv1 domain.
Shaded regions indicate ±1 standard deviation.





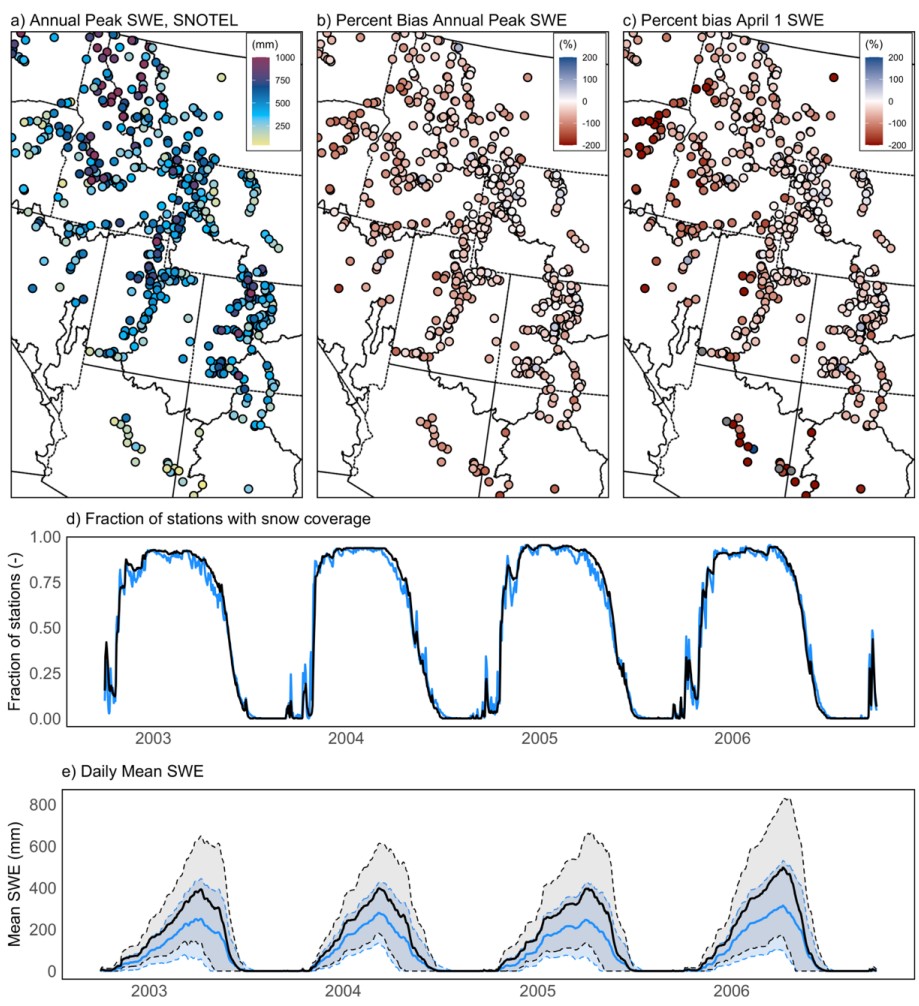

Figure 10: Summary of PFCONUSv1 modeled snow water equivalent performance relative to SNOTEL sites. Shown are observed peak SWE at SNOTEL sites (a), percent bias for peak SWE (b) and April 1 SWE (c), daily spatial fraction of stations with snow coverage (d) and mean daily SWE (e). In (e), shaded regions indicate ±1 standard deviation in space.


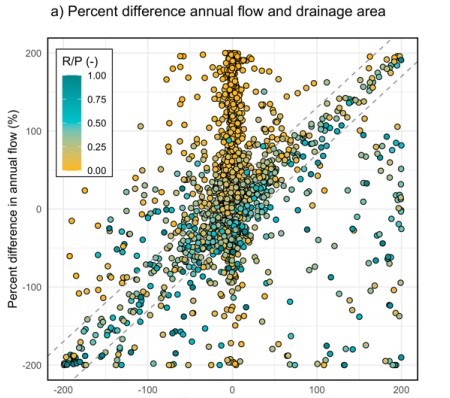

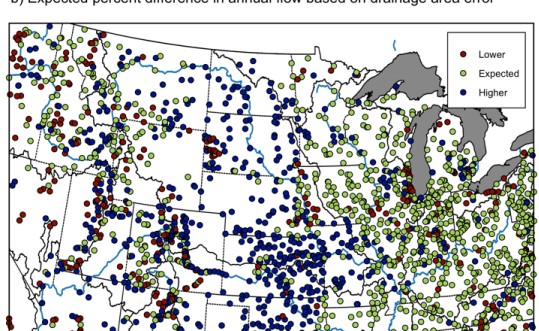

Figure 11: a) Percent difference between observed and simulated annual flow volume as a function of percent
difference in true and PFCONUSv1 drainage area, colored by annual runoff ratio. b) Locations where error in
simulated flow volume is greater than, less than, or expected from drainage area bias. *Expected* behavior was
defined as locations that lie within the ±30% dashed error bars shown in (a).



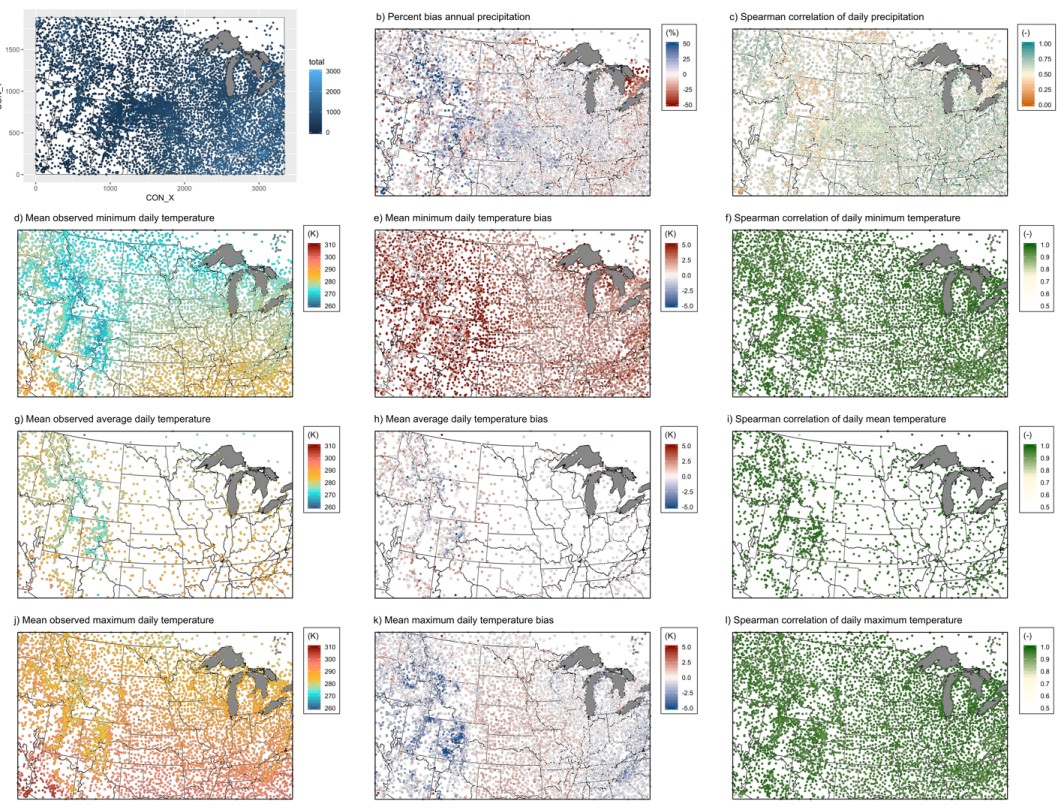

Figure 12: Observed precipitation and temperature at GHCND meteorological stations compared to interpolated NLDAS at their nearest neighbor PFCONUSv1 cell. a) Observed cumulative annual precipitation, b) percent bias in annual precipitation, c) Spearman's $\rho$ between simulated and observed daily precipitation. Also shown are observed average daily minimum (d), average (g), and maximum (j) temperature, the total bias in minimum (e), average (h), and maximum daily temperature (k), and the Spearman correlation for minimum (f), average (i) and maximum (l) daily temperature.


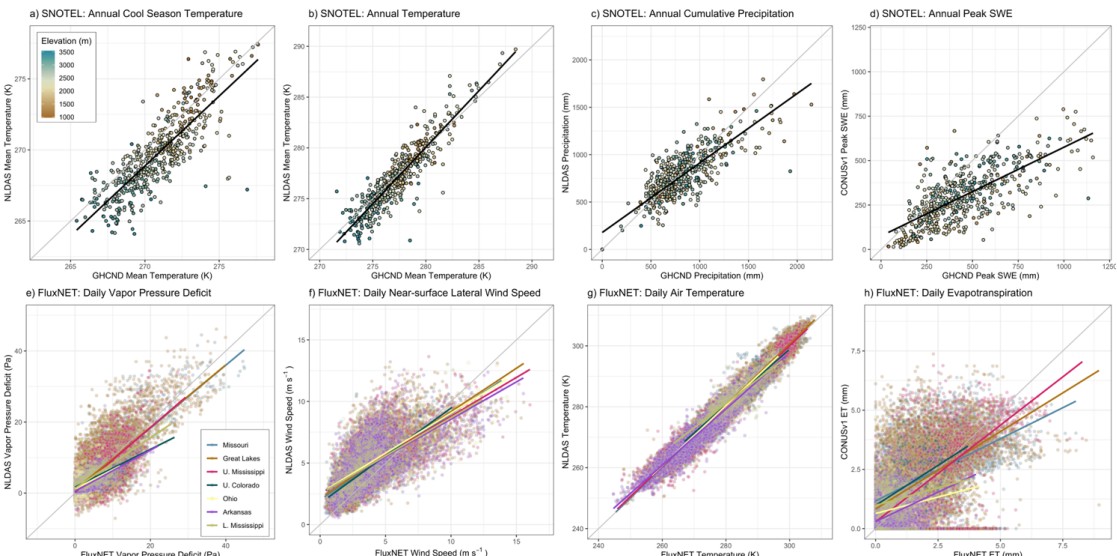

Figure 13: Meteorological forcing, SWE and ET bias at SNOTEL and FLUXNET stations. At SNOTEL locations, colored by elevation: a) mean NLDAS mean cool season temperature versus observed cool season temperature, b) mean NLDAS annual temperature versus observed cool season temperature, c) NLDAS annual cumulative precipitation versus observed, and d) PFCONUSv1 annual peak SWE versus observed. At FLUXNET locations, colored by the major basin location of the FLUXNET site: e) Daily NLDAS vapor pressure deficit versus observed,
f) daily NLDAS near-surface lateral wind speed versus observed, g) daily NLDAS mean air temperature versus observed, and h) PFCONUSv1 daily ET versus observed. Lines show linear regression with $p<0.05$ in all cases.



**Code and data availability**

ParFlow-CLM is an open-source, parallel, modular hydrologic model that is freely available on Github at
https://github.com/parflow/parflow.git. All data generated from the ParFlow-CLM CONUS configuration version

1.0 is available upon request. Given the considerable storage demand (approximately 60 terabytes for four water
years of hourly data, including forcing and daily or monthly processed climatologies), the model outputs are stored
on a private server. The authors will coordinate with the HydroFrame project team, funded through the NSF
Cyberinfrastructure for Sustained Scientific Innovation (CSSI) project, to ultimately provide a FAIR-aligned,
publicly accessible data repository of all raw model results. A primary objective of the HydroFrame project is to

provide a platform for users to freely access PFCONUS model results, as well as to subset or modify inputs and
forcing to locally run their own ParFlow-CLM simulations. As HydroFrame capabilities develop and future versions
are completed, we plan to make PFCONUS results publicly available through this platform.

**Author contribution**

Dr. Mary O'Neill prepared and ran PFCONUSv1 simulations, processed model outputs and analyzed model
performance, and prepared the manuscript with contributions from coauthors. Dr. Reed Maxwell and Dr. Laura
Condon provided some processing codes for spatiotemporal averaging, offered guidance for interpreting integrated
hydrologic simulations and managing data, and aided in manuscript preparation and editing. Danielle Tijerina
contributed to manuscript preparation, editing and review.


**Competing interests**

The authors declare that they have no conflict of interest.

**Acknowledgements**

The authors gratefully acknowledge funding, provided in part by the U.S. Department of Energy, Interoperable
Design of Extreme-scale Application Software (IDEAS) Project, and in part by the National Science Foundation
Office of Advanced Cyberinfrastructure, Cyberinfrastructure for Sustained Scientific Innovation (CSSI) project
under Award Number 1835903. High performance computing resources were provided by the National Center for
Atmospheric Research (NCAR) Computational and Information Systems Laboratory (CISL) Cheyenne computing

platform.