# Peer review of "Assessment of the ParFlow-CLM CONUS 1.0 integrated hydrologic model: Evaluation of hyper-resolution water balance components across the contiguous United States"

_Geoscientific Model Development, 2020_

## Short Comment (SC1) · 21 Dec 2020

Dear authors,

in my role as Executive editor of GMD, I would like to bring to your attention our Editorial version 1.2:

https://www.geosci-model-dev.net/12/2215/2019/

This highlights some requirements of papers published in GMD, which is also available

on the GMD website in the 'Manuscript Types' section: http://www.geoscientific-model-development.net/submission/manuscript_types.html

In particular, please note that for your paper, the following requirement has not been met in the Discussions paper:

- Code must be published on a persistent public archive with a unique identifier for the exact model version described in the paper or uploaded to the supplement, unless this is impossible for reasons beyond the control of authors. All papers must include a section, at the end of the paper, entitled "Code availability". Here, either instructions for obtaining the code, or the reasons why the code is not available should be clearly stated. It is preferred for the code to be uploaded as a supplement or to be made available at a data repository with an associated DOI (digital object identifier) for the exact model version described in the paper. Alternatively, for established models, there may be an existing means of accessing the code through a particular system. In this case, there must exist a means of permanently accessing the precise model version described in the paper. In some cases, authors may prefer to put models on their own website, or to act as a point of contact for obtaining the code. Given the impermanence of websites and email addresses, this is not encouraged, and authors should consider improving the availability with a more permanent arrangement. Making code available through personal websites or via email contact to the authors is not sufficient. After the paper is accepted the model archive should be updated to include a link to the GMD paper.

As GitHub is not a persistent archive, please provide a persistent release for the exact source code version used for the publication in this paper. As explained in https://www.geoscientific-model-development.net/about/manuscript_types.html the preferred reference to this release is through the use of a DOI which then can be cited in the paper. For projects in GitHub a DOI for a released code version can easily be

**[GMDD](GMDD)**

created using Zenodo, see https://guides.github.com/activities/citable-code/ for details.

Yours, Astrid Kerkweg

---

## Referee Comment (RC1) · Anonymous Referee #1 · 6 Apr 2021

Authors performed a multi-criteria evaluation of a continental scale integrated hydrologic model to identify sources of bias and error in hydrologic model predictions. While many continental scale hydrologic and land surface models have been developed and validated against observations, none of them have offered such a comprehensive evaluation using multiple point based and spatially distributed observations. Furthermore, many global scale models employ empirical formulations for simulating hydrologic processes to reduce computational time. Therefore, continental scale evaluation of a physically based model like PFCONUS is new. The study is comprehensive and very well written and organized. While model simulations for a longer time frame would be desired, I am aware of the scale of simulations and analysis done by the authors and only recommend a few minor comments below:

1) Add units of variables defined for each equation.

2) Line 465- Add a label to Figure 3 to identify locations/extent of different river basins or label the basins in Figure 2.

3) Could you please add NSE metric to Figure 4

4) Line 540 – Could you please compare performance of MOD16A2 and SSEBop against FLUXNET data using the same period as PFCONUSv1?

5) Line 561- Correct subplot number. In Figure 6, final row has the same subplot number as the previous row.

6) Line 567- Correct Figure number. It should be Fig. 6

7) Line 690- Update figure number to Fig. 8d

8) For soil moisture comparison- Did you compare top layer simulated soil moisture with the ESA dataset?

9) Line 1325- ESACCI does not have mascon solution.

10) In Figure 9 – Does shaded region show standard deviation of spatially distributed soil moisture?

11) Line 824- Change Fig 7g.h to Fig 6 g.h

12) Line 841- Add "temperature"

13) Line 844-845- It is not clear. Please clarify.

14) For discussion of uncertainty in meteorological forcing using

ParFlow-CLM, I refer authors to Schreiner-McGraw and Ajami (2020, https://doi.org/10.1029/2020WR027639)

15) Figure 10- Add legend to subplots d and e

16) Figure 12 – Why the density of GHCND gauges are smaller in 12g-I compared to 12 j,k,l

---

## Referee Comment (RC2) · Anonymous Referee #2 · 18 Apr 2021

The manuscript presents an evaluation of a continental-scale, hyper-resolution model run of a fully coupled hydrologic model. The presented results may serve as a benchmark for model inter-comparisons and guide future ParFlow model improvements. The manuscript is polished and detailed. This reviewer is certainly impressed by its exhaustiveness. Before the manuscript can be recommended for publication in the as-is form, authors are encouraged to address the following concerns:

a) Argument should be provided to explain why only 30 FluXNet sites were chosen?

What were the criteria for selecting them (e.g., belonging to different PFTs or hydroclimatic regions)?

b) Throughout the manuscript, subjective qualifiers have been added to describe the ability of the model to replicate observations. These include "appropriately", "exceptionally well", "good performance", "acceptable", "poor", "very good ability", "moderate to strong" etc. However, it is not clear for a (future) model user if the results are indeed appropriate (and for what?), exceptionally good (for what?), or acceptable (for what and when), etc. Clear guidance should be provided for decoding these qualifiers, especially as a single qualifier is sometimes used for a range of variables, even when the accuracy is very different. Should these qualifiers be interpreted only for qualitative inter-comparison across sites or are these describing relative performance w.r.t. the performance of other models? To this reader, subjective qualifiers encourage a biased evaluation.

C) While the presented runoff comparisons are quite exhaustive, (future)users of the model and other readers will benefit if exceedance plots for performance metrics are generated. Such a plot will highlight the fraction of observation locations with performance higher than a threshold. These plots may also be generated for different river basins to highlight the relative performance between them. It will help the regional scale modelers. Finally, as it is mentioned that large errors could have been introduced by anthropogenic impacts such as dams etc., separate performance comparisons in Gages-II watersheds could be performed to help address this point. Similar exceedance plots should be generated for other variables as well, as these plots highlight how many grids show performance better than a given threshold, and thus underscore the usefulness of spatially explicit modeling.

D)The uniqueness of this model is its integrated, fully-coupled nature (and its application at hyper-resolution over continental scale). However, none of the comparisons highlight how usage of this kind of model can provide better estimates in at least some variables at certain locations, over other LSMs. Authors are encouraged to consider

showing some relevant comparisons along these lines to significantly increase the impact of the paper.

E) Conclusions focus on highlighting that the model produces good temporal patterns. What is not clear is if one needs such a complex, fully distributed coupled model to obtain comparable correlations. Most temporal correlations in hydrologic responses are largely driven by the temporality of precipitation/melt and Rn/Temperature/VPD. Unless it can be shown the subsurface-surface flow interactions, a "differentiator" characteristic of this model w.r.t. LSMs, has led to improvement in these temporal correlations, demonstration of the real efficacy of this model (considering its data and computational demands) w.r.t. LSMs remains unclear.

Minor comment:

– Southeastern states are cut out from the figures. Interested (future) users of the model in these states will find it difficult to follow through. – Line 248: "Stave IV" should be "Stage IV". – Line 415: "shear" should be "sheer" – Line 454: Fig. 3.3e should be 3e. – Line 567: Fig. 3.6 should be Fig. 6. – Line 597: Figure 6 should be Figure 7 – Fig. 12a: The color palette does not highlight the heterogeneity well. Please consider alternatives.

---

## Author Comment (AC1) · 30 Jun 2021

Authors performed a multi-criteria evaluation of a continental scale integrated hydrologic model to identify sources of bias and error in hydrologic model predictions. While many continental scale hydrologic and land surface models have been developed and validated against observations, none of them have offered such a comprehensive evaluation using multiple point based and spatially distributed observations. Furthermore, many global scale models employ empirical formulations for simulating hydrologic processes to reduce computational time. Therefore, continental scale evaluation of a physically based model like PFCONUS is new. The study is comprehensive and very well written and organized. While model simulations for a longer time frame would be desired, I am aware of the scale of simulations and analysis done by the authors and only recommend a few minor comments below:

We thank the referee for their consideration, time, and thorough read of our manuscript. We have replied to your comments in blue text below. In some cases, we show text that has been added to or replaced sections of the original manuscript. The new additions or revised text are shown in green.

1) Add units of variables defined for each equation.

Dimensions have been added to physical equations (1) through (7).

2) Line 465- Add a label to Figure 3 to identify locations/extent of different river basins or label the basins in Figure 2.

Major basins have been labeled in Figure 2.

3) Could you please add NSE metric to Figure 4

Nash-Sutcliffe Efficiency was originally left out given some of the known problems with NSE as a performance criterion (Gupta et al., 2009). However, we recognize that the metric is widely used and that some readers and regional scale modelers would benefit from its addition. Given the manuscript's length, we have decided to add NSE in a supplemental text to the manuscript; we have also added the Kling-Gupta Efficiency (KGE, Gupta et al., 2009; Kling et al., 2012) in the supplemental information, as it addresses some of NSE's deficiencies and combines variability, bias, and correlation (Kling et al., 2012). The corresponding figure reporting NSE and KGE error metrics for PFCONUSv1 simulated daily streamflow is shown below (Figure S9).

a) Kling-Gupta Efficiency of Daily Flow

[Figure]

b) Nash-Sutcliffe Efficiency of Daily Flow

[Figure]

Figure S9: KGE (a) and NSE (b) evaluated at USGS stream gauges for PFCONUSv1 simulated daily streamflow.

4) Line 540 – Could you please compare performance of MOD16A2 and SSEBop against FLUXNET data using the same period as PFCONUSv1?

While evaluation of MOD16A2 and SSEBop algorithms with respect to FLUXNET observations is certainly valuable, we prefer to use the MODIS products at an aggregated temporal and spatial scale to evaluate PFCONUSv1 model results. As detailed in line 310 in the manuscript, there exist considerable uncertainties in point-scale MODIS values and high temporal resolutions associated with cloud cover and other limitations. We refer the reviewer to Velpuri et al. (2013) and Westerhoff (2015) for detailed evaluation of MODIS application at small spatial scales, and we feel that an adequate validation of the two MODIS algorithms against FLUXNET for the simulation period is outside the scope of this paper.

5) Line 561- Correct subplot number. In Figure 6, final row has the same subplot number as the previous row.

Thank you, this has been fixed.

6) Line 567- Correct Figure number. It should be Fig. 6

Thank you, fixed.

7) Line 690- Update figure number to Fig. 8d

Thank you, fixed.

8) For soil moisture comparison- Did you compare top layer simulated soil moisture with the ESA dataset?

Yes, only the top layer. This clarification has been added to the beginning of section 3.4.2, as well as to section 2.3.3.

9) Line 1325- ESACCI does not have mascon solution.

This typo has been fixed.

10) In Figure 9 – Does shaded region show standard deviation of spatially distributed soil moisture?

Yes, standard deviation was taken spatially across the major basin. This has been clarified in the figure caption for Figure 9.

11) Line 824- Change Fig 7g.h to Fig 6 g.h

Thank you, this has been fixed.

12) Line 841- Add "temperature"

Added.

13) Line 844-845- It is not clear. Please clarify.

The discussion here is referring to the fact that, on average and across all FLUXNET sites, ParFlow-CLM over (under) estimates low (high) rates of daily ET (Figure 13h) relative to observations. We have changed the text to clarify the point. This section now reads:

> "Overall, PFCONUSv1 under(over)-estimates relatively high (low) daily evapotranspiration rates (Figure 13h). For FLUXNET locations and days exhibiting ET rates over (under) 4 mm day$^{-1}$, mean daily bias is -1.2 mm (0.3 mm). Biases in NLDAS vapor pressure and wind speed could be a contributing factor. Lower vapor pressure deficits (0 to 20 Pa) and lower wind speeds (0 to 6 m s$^{-1}$) have an overall positive bias, which could explain PFCONUSv1 overpredicting low ET days. Similarly, we believe the bias on high-evapotranspiration days (ET > 4 mm day$^{-1}$), which PFCONUSv1 preferentially under-predicts, may be attributed to NLDAS under predicting wind speeds greater than ~10 m s$^{-1}$."

14) For discussion of uncertainty in meteorological forcing using ParFlow-CLM, I refer authors to Schreiner-McGraw and Ajami (2020, https://doi.org/10.1029/2020WR027639)

Thank you, we have included this reference in our discussion.

15) Figure 10- Add legend to subplots d and e

Legend showing the model simulation as blue lines/shading and the observed SNOTEL values as black lines/shading has been added to subplot e.

16) Figure 12 – Why the density of GHCND gauges are smaller in 12g-I compared to 12 j,k,l

Average daily temperature was taken directly from the GHCND network at meteorological stations with that metric available. Daily maximum and minimum temperatures were available at more GHCN stations than daily mean temperature; the GHCN network for mean daily temperature was simply less dense. We could in theory approximate mean temperature using the maximum and minimum values at more sites, but we believe this would defeat the purpose of using a quality-controlled network of observations to compare to our model results. The number of GHCN sites with available daily max/min and daily mean are detailed in the first paragraph of section 2.3.4.

References

Gupta, H. V., Kling, H., Yilmaz, K. K., & Martinez, G. F. (2009). Decomposition of the mean squared error and NSE performance criteria: Implications for improving hydrological modelling. *Journal of Hydrology*, 370(1–2), 80–91.

Kling, H., Fuchs, M., & Paulin, M. (2012). Runoff conditions in the upper Danube basin under an ensemble of climate change scenarios. *Journal of Hydrology*, 424–425, 264–277.

Velpuri, N. M., Senay, G. B., Singh, R. K., Bohms, S., & Verdin, J. P. (2013). A comprehensive evaluation of two MODIS evapotranspiration products over the conterminous United States: Using point and gridded FLUXNET and water balance ET. *Remote Sensing of Environment*, *139*, 35–49. https://doi.org/10.1016/j.rse.2013.07.013

Westerhoff, R. S. (2015). Using uncertainty of Penman and Penman-Monteith methods in combined satellite and ground-based evapotranspiration estimates. *Remote Sensing of Environment*, *169*, 102–112. https://doi.org/10.1016/j.rse.2015.07.021

---

## Author Comment (AC2) · 30 Jun 2021

The manuscript presents an evaluation of a continental-scale, hyper-resolution model run of a fully coupled hydrologic model. The presented results may serve as a benchmark for model inter-comparisons and guide future ParFlow model improvements. The manuscript is polished and detailed. This reviewer is certainly impressed by its exhaustiveness. Before the manuscript can be recommended for publication in the as-is form, authors are encouraged to address the following concerns:

We are very grateful to this referee for their thoughtful critique and suggestions, which we believe have considerably added to the quality of the manuscript. We have replied to your comments in blue text below. In some cases, we show text that has been added to or replaced sections of the original manuscript. The new additions or revised text are shown in green.

a) Argument should be provided to explain why only 30 FluXNet sites were chosen? C1 What were the criteria for selecting them (e.g., belonging to different PFTs or hydroclimatic regions)?

We used all available FLUXNET locations with at least one water year of overlap with our simulation period – i.e., those that contained one or more years of observations between October 1, 2002, and September 30, 2006. This description has been clarified in section 2.3.2 of the manuscript. The text now reads:

> "FLUXNET data were obtained from the FLUXNET 2015 online data portal (https://fluxnet.fluxdata.org/, accessed February 6, 2020), and the 30 sites used in this study are those that contain at least one water year of observations during the simulation period."

b) Throughout the manuscript, subjective qualifiers have been added to describe the ability of the model to replicate observations. These include "appropriately", "exceptionally well", "good performance", "acceptable", "poor", "very good ability", "moderate to strong" etc. However, it is not clear for a (future) model user if the results are indeed appropriate (and for what?), exceptionally good (for what?), or acceptable (for what and when), etc. Clear guidance should be provided for decoding these qualifiers, especially as a single qualifier is sometimes used for a range of variables, even when the accuracy is very different. Should these qualifiers be interpreted only for qualitative inter-comparison across sites or are these describing relative performance w.r.t. the performance of other models? To this reader, subjective qualifiers encourage a biased evaluation.

We thank the reviewer for their input, and we want the evaluation of our model to be presented with as little bias as possible. There are three ways in which we have addressed these concerns:

1) The purpose of and recommended interpretation for the performance metrics used in our study has been clarified with the addition of the following text, at the end of section 2.4 (Performance metrics):

"Together, performance metrics (8) through (10) are quantitative indicators of model realism, representing a model's ability to capture long term states (PBIAS), timing ($\rho$), and variability (RSR). However, many other statistical criteria are popular (Waseem et al., 2017), and the target values used to indicate unacceptable, acceptable, or excellent performance can vary because criteria for evaluation necessarily depend upon model purpose (i.e., a regional surface water model that has been well calibrated for operational forecasting will represent spatiotemporal patterns of streamflow with higher accuracy than a continental-scale land surface model can plausibly achieve). Further, performance is expected to decrease with increasingly higher temporal resolution: For instance, criteria may be more lenient across all error metrics when moving from monthly to daily timescales at the watershed scale (Moriasi et al., 2015) as well as from seasonal to monthly timescales at the global scale (Krysanova et al,. 2020). As a physically-based, high-resolution (spatially and temporally) and uncalibrated continental-scale model, a primary purpose of the PFCONUS, and others like it (Gleeson et al., 2021), is to understand process interactions between groundwater, surface-water, and ecohydrological fluxes. In this study, a PFCONUS simulated water balance component in (2) is judged to be excellent for this purpose with the following measures: RSR<0.6, $\rho$>0.7, or |PBIAS|<20%. Locations that indicate unacceptable or poor performance are those with RSR<1.2, |PBIAS|<75%, and $\rho$>0.5. However, error metrics are reported with the primary goal of inter-comparison across locations (interpretation of metrics should be paired with visual inspection of spatial patterns and timeseries provided), or, where discussed, relative to the performance of other continental-scale hydrologic or land surface models. Gleeson et al. (2021) caution against the use of model evaluation to indicate a "finished" product, and instead recommend open-ended evaluation and model improvement. Metrics (8) through (10) are therefore used here to identify where future development of PFCONUS can be focused to improve upon timing, volume, and variability of fluxes. Performance metrics reported in this study are also supplemented by plots of probability of exceedance or non-exceedance where appropriate (see the Supplemental Information, Figures S1 through S8), which should help regional scale modelers identify relative performance at various thresholds. Since there exist many other commonly used performance metrics particular to streamflow, we also report Nash-Sutcliff Efficiency and Kling-Gupta Efficiency for simulated flows at USGS gauges (Figure S9 in the supplemental text) (Gupta et al., 2009)."

2) We have made every effort to remove or rephrase subjective qualifiers such as those listed by the referee. An example is the paragraph (beginning at line 452 of the original manuscript) reporting the error statistics for PFCONUSv1 streamflow compared to USGS stream gauges. The original version of the manuscript classified modeled streamflow performance as "excellent", "very good", "good", "acceptable", and "poor"; the new text, shown below, omits all these subjective descriptors, while still reporting the overall distribution of streamflow performance, which was the primary purpose of this section. There are many other instances in the text in which we have removed subjective language entirely.

Line 452: "PFCONUSv1 reproduces point-scale annual flows across the United States with a median annual PBIAS of 7.7 %, and with 25th and 75th percentiles of -26.2% and 77.4%, respectively (Fig. 3b). Shown in Fig. 3e and f, the 25th, 50th, and 75th percentiles for daily Spearman's $\rho$ are 0.42, 0.65, and 0.76, while the same for RSR are 0.86, 1.2, and 2.5. The median PFCONUSv1 minus USGS difference in RR is 0.016 (Fig. 3d), which corresponds to a mean percent bias in runoff ratio of 8.3%. The PFCONUSv1 model simulates observed streamflow with RSR<0.6, $\rho$>0.7, and |PBIAS|<20% at 54 gauges (approximately 2% of available sites). An additional 97 locations (4% of gauges) exhibit RSR<0.7, $\rho$>0.65, and |PBIAS|<30%. An additional 382 locations (15.7% of gauges) show RSR<1, $\rho$>0.6, and |PBIAS|<50%. And, finally, and an additional 268 gauges (11% of gauges) show RSR<1.2, $\rho$>0.5, and |PBIAS|<75%. As has been shown in previous literature (Waseem et al., 2017), different performance metrics do not always indicate the same closeness of fit: While 2099 gauges (86% of the dataset) show either RSR<1.2, |PBIAS|<75%, or $\rho$>0.5, only 801 gauges (34% of all gauges) fit all those criteria."

3)  A supplementary file is now available with probability of exceedance plots (which also address the referee's point C), so that the full distribution of performance across space is transparent, and in order to allow modelers to determine if, e.g., the streamflow performance of the model meets their own performance requirements for a given major basin.

C) While the presented runoff comparisons are quite exhaustive, (future)users of the model and other readers will benefit if exceedance plots for performance metrics are generated. Such a plot will highlight the fraction of observation locations with performance higher than a threshold. These plots may also be generated for different river basins to highlight the relative performance between them. It will help the regional scale modelers. Finally, as it is mentioned that large errors could have been introduced by anthropogenic impacts such as dams etc., separate performance comparisons in Gages-II watersheds could be performed to help address this point. Similar exceedance plots should be generated for other variables as well, as these plots highlight how many grids show performance better than a given threshold, and thus underscore the usefulness of spatially explicit modeling.

Thank you for the suggestion; we have found it to be very helpful, especially in demonstrating the overall performance at the GAGES-II reference gauge locations, versus other sites that are subject to higher anthropogenic influence. Given the current manuscript length and extensive comparisons already made in the form of maps and timeseries, we have decided to include these plots in a supplemental text. We specifically draw this referee's attention to Figure S1 and S2, shown below, which we believe best address the referee's point. Figures S1 and S2 show probability of exceedance (or non-exceedance) for various streamflow performance metrics, reference vs non-reference gauges, and major basins. The supplemental text will also include probability of exceedance for other point scale observations and for MODIS products. Since the ESACCI and GRACE remote sensing data were aggregated to the major basin scale for comparison to PFCONUSv1, exceedance plots were not generated for these products as there were too few comparison points (major basins).

[Figure]

Figure S1: Probability of exceedance for a) Spearman's $\rho$, b) $R^2$, c) NSE, and d) KGE, and probability of non-exceedance for e) the absolute relative bias (the absolute value of percent bias expressed as a decimal) and f) RSR, for PFCONUSv1 simulated daily streamflow compared to USGS stream gauges. Results are shown for both non-reference (black) and reference (red) gauges based on the classification detailed in Maxwell et al. (2015), which distinguishes reference gauges to be those with least anthropogenic influence (groundwater abstractions, dams, diversions, etc.) to their upstream area based on the GAGES-II dataset.

[Figure]

Figure S2: Same as Figure S1, colored by major basin rather than by reference and non-reference locations.

D)The uniqueness of this model is its integrated, fully-coupled nature (and its application at hyper-resolution over continental scale). However, none of the comparisons highlight how usage of this kind of model can provide better estimates in at least some variables at certain locations, over other LSMs. Authors are encouraged to consider showing some relevant comparisons along these lines to significantly increase the impact of the paper.

E) Conclusions focus on highlighting that the model produces good temporal patterns. What is not clear is if one needs such a complex, fully distributed coupled model to obtain comparable correlations. Most temporal correlations in hydrologic responses are largely driven by the temporality of precipitation/melt and Rn/Temperature/VPD. Unless it can be shown the subsurface-surface flow interactions, a "differentiator" characteristic of this model w.r.t. LSMs, has led to improvement in these temporal correlations, demonstration of the real efficacy of this model (considering its data and computational demands) w.r.t. LSMs remains unclear.

Points D) and E) are addressed here, together, since they both introduce a common point – that the metrics presented in this manuscript would be more valuable if relevant comparisons to LSMs (and perhaps other global hydrologic models with various levels of groundwater representation) were conducted.

First, as a minor point, the first paragraph of the introduction is no longer limited to summarizing only Spearman's rho performance. We have added summaries in that paragraph of |PBIAS| and RSR to outline our model's ability to reproduce large scale magnitude and error to variability ratio of fluxes, respectively. Second, we firmly agree with the reviewer that the benefits of an in-depth comparison to LSMs would be immeasurable. Indeed, inter-model comparison projects have recently been suggested as a major resource to continental and global-scale model evaluation (with or without the incorporation of point-based or remote sensing observations as reference) (Gleeson et al., 2021). Unfortunately, an adequate, detailed model comparison study is simply outside the scope of this already extensive manuscript. Also, we believe there is great value in evaluating this model's performance, even without comparing its performance to that of other models. This evaluation (without model comparison) still allows for intercomparison of model performance across space and time, which can help highlight problem areas, attribute bias sources, and encourage iterative model improvement. Further, work has already begun to formally evaluate PFCONUSv1 performance relative to that of other models – specifically, in the phase 1 of the Continental Hydrology Intercomparison Project (CHIP) (Tijerina et al., 2021), which compares simulated streamflow performance of PFCONUSv1 to that of the National Water Model configuration of the WRF-Hydro hydrologic model. A goal is to have our model continue to be incorporated into current and future model evaluation and comparison platforms.

Minor comment:

– Southeastern states are cut out from the figures. Interested (future) users of the model in these states will find it difficult to follow through.

This first version of the PFCONUS model extends only as far as the boxed outline in Figure 2 (and subsequent figures). This model follows the original ParFlow model (uncoupled to CLM) which was designed and configured over the contiguous U.S. and described in Maxwell et al., 2015. While current published PFCONUS results are limited to this box domain, there is a considerable effort to extend PFCONUS to the coastlines and to the edges of major basin boundaries in Canada and Mexico like the Columbia and Rio Grande (e.g., extending aquifer parameterization by taking advantage of novel, hyper-resolution hydrogeological data; de Graaf, 2020). We hope that future versions that include these states will (soon) be available to future users.

– Line 248: "Stave IV" should be "Stage IV".

Thank you, this has been corrected.

– Line 415: "shear" should be "sheer"

Corrected.

– Line 454: Fig. 3.3e should be 3e.

Corrected.

– Line 567: Fig. 3.6 should be Fig. 6.

Corrected.

– Line 597: Figure 6 should be Figure 7

Corrected.

– Fig. 12a: The color palette does not highlight the heterogeneity well. Please consider alternatives.

Thank you for pointing this out. It was a plot formatting mistake – the new Figure 12 shows the correct color scale with subplot (a) adjusted to mirror sizing and scale of other subplots. This is shown below.

[Figure]

Figure 12: Observed precipitation and temperature at GHCND meteorological stations compared to interpolated NLDAS at their nearest neighbor PFCONUSv1 cell. a) Observed cumulative annual precipitation, b) percent bias in annual precipitation, c) Spearman's $\rho$ between simulated and observed daily precipitation. Also shown are observed average daily minimum (d), average (g), and maximum (j) temperature, the total bias in minimum (e), average (h), and maximum daily temperature (k), and the Spearman correlation for minimum (f), average (i) and maximum (l) daily temperature.

References

de Graaf, I., Condon, L., & Maxwell, R. (2020). Hyper-resolution continental-scale 3-D aquifer parameterization for groundwater modeling. Water Resources Research, 56, e2019WR026004. https://doi.org/10.1029/2019WR026004

Gleeson, T., Wagener, T., Döll, P., Zipper, S. C., West, C., Wada, Y., Taylor, R., Scanlon, B., Rosolem, R., Rahman, S., Oshinlaja, N., Maxwell, R., Lo, M.-H., Kim, H., Hill, M., Hartmann, A., Fogg, G., Famiglietti, J. S., Ducharne, A., de Graaf, I., Cuthbert, M., Condon, L., Bresciani, E., and Bierkens, M. F. P.: GMD Perspective: the quest to improve the evaluation of groundwater representation in continental to global scale models, Geosci. Model Dev. Discuss. [preprint], https://doi.org/10.5194/gmd-2021-97, in review, 2021.

Krysanova, V., Zaherpour, J., Didovets, I., Gosling, S.N., Gerten, D., Hanasaki, N., Schmied, H.M., Pokhrel, Y., Satoh, Y., Tang, Q., & Wada, Y. (2020). How Evaluation of Global Hydrological Models Can Help to Improve Credibility of River Discharge Projections under Climate Change. *Climatic Change*, *163*, 1353-1377. https://doi.org/10.1007/s10584-020-02840-0

Maxwell, R. M., Condon, L. E., & Kollet, S. J. (2015). A high-resolution simulation of groundwater and surface water over most of the continental US with the integrated hydrologic model ParFlow v3. *Geoscientific Model Development*, *8*(3), 923–937. https://doi.org/10.5194/gmd-8-923-2015

Moriasi, D.N., Gitau, M.W., Pai, N., & Daggupati, P. (2015). Hydrologic and Water Quality Models: Performance Measures and Evaluation Criteria. *Transactions of the ASABE*, *58*(6), 1763-1785. DOI 10.13031/trans.58.10715

Tijerina, D.T., Condon, L.E., FitzGerald, K., Dugger, A., O'Neill, M.M., Sampson, K., Gochis, D.J., Maxwell, R.M. (2021). Continental Hydrologic Intercomparison Project (CHIP), Phase 1: A Large-Scale Hydrologic Model Comparison over the Continental United States. *Water Resources Research*, *in press*.

---

## Author Comment (AC3) · 1 Jul 2021

Thank you very much for the clarification. ParFlow-CLM v3.6 is now archived on Zenodo: https://doi.org/10.5281/zenodo.4639760. We will update the code and data availability section in the manuscript accordingly.